# Pay Attention to Small Weights

**Chao Zhou**    **Tom Jacobs**    **Advait Gadhikar**    **Rebekka Burkholz**
CISPA Helmholtz Center for Information Security, Saarbrücken, Germany
{chao.zhou, tom.jacobs, advait.gadhikar, burkholz}@cispa.de

## Abstract

Finetuning large pretrained neural networks is known to be resource-intensive, both in terms of memory and computational cost. To mitigate this, a common approach is to restrict training to a subset of the model parameters. By analyzing the relationship between gradients and weights during finetuning, we observe a notable pattern: large gradients are often associated with small-magnitude weights. This correlation is more pronounced in finetuning settings than in training from scratch. Motivated by this observation, we propose NANOADAM, which dynamically updates only the small-magnitude weights during finetuning and offers several practical advantages: first, the criterion is *gradient-free*—the parameter subset can be determined without gradient computation; second, it preserves large-magnitude weights, which are likely to encode critical features learned during pretraining, thereby reducing the risk of catastrophic forgetting; thirdly, it permits the use of larger learning rates and consistently leads to better generalization performance in experiments. We demonstrate this for both NLP and vision tasks.

## 1   Introduction

With the advent of Transformer-based models like GPT [26], large models (LMs) [32, 20] excel across domains such as natural language processing (NLP) [3, 37] and computer vision (CV) [21, 41]. They enable effective knowledge transfer via finetuning (FT) on downstream tasks, facilitating the development of domain-specific models.

It is well known that fully finetuning LMs requires substantial computations and memory [33, 12]. One of the main reasons is that the predominant optimizers, Adam [16] and its variants [22], maintain both first- and second-order momentum estimates for each parameter [16, 22]. For a model with $N$ trainable parameters, this results in a memory footprint equivalent to storing approximately $3N$ parameters, significantly limiting scalability. To address this problem, methods such as gradient checkpointing [4], quantization [11], and parameter offloading [28] have been developed. Gradient checkpointing [4], for instance, reduces memory usage by storing intermediate feature maps and gradients and recomputing them during backpropagation—a trade-off that sacrifices computational efficiency for reduced memory demand. Similarly, 8-bit Adam [7] addresses memory overhead by quantizing optimizer statistics to 8-bit precision while leveraging block-wise dynamic quantization to maintain numerical stability, thereby minimizing performance degradation.

Recently, projection-based methods have emerged as a promising approach to reduce memory overhead. For instance, GaLore [44] enables full-parameter training by projecting gradients into a low-rank subspace, applying Adam-like updates there, and projecting them back to the original space—thus reducing memory usage. However, GaLore relies on SVD decomposition of gradients, which is only applicable to layers satisfying the reversibility property [27]. MicroAdam [24] compresses gradients using top-$k$ selection and mitigates performance loss via an error feedback mechanism inspired by distributed training. However, it still incurs memory overhead for storing accumulated error and introduces extra computation for quantization and feedback. In contrast, our

method does not require gradient information to determine which parameters to update and eliminates the need for error feedback, resulting in improved efficiency.

**Contributions**   In this paper, we investigate the relationship between gradients and parameter magnitudes in the context of finetuning LMs. Our empirical analysis reveals a strong pattern: parameters with large gradients are associated with small magnitudes. While this correlation is not perfect—smallest weights do not strictly correspond to largest gradients—we find that selectively updating small-weight parameters is consistently more effective than updating those with large gradients. Yet, learning small weights works because of the association with large gradients that support learning. To deepen this insight, we provide a theoretical analysis using a two-layer teacher–student framework, showing that updating small weights not only yields more efficient learning but also helps mitigate catastrophic forgetting during finetuning. Conceptually, our analysis reconciles two distinct sparse finetuning principles: training parameters with large gradients versus small magnitude, showing that they act in tandem. While the former is the more dominant principle, we systematically highlight the advantages of the latter.

Motivated by these findings, we propose NANOADAM, an optimizer that selectively updates parameters with small absolute magnitudes. Compared to prior methods, NANOADAM offers several advantages: (1) it avoids reliance on gradient information, allowing precomputation of update masks and dynamic sparsity control; (2) it eliminates the need for error feedback, improving both memory and computational efficiency compared to microAdam [24]; (3) after finetuning, over 80% of large parameters remain untouched, depending on the sparsity level; and (4) by avoiding updates to large weights, it preserves critical features from pretraining, mitigating catastrophic forgetting [17]. Additionally, by leaving large weights unchanged, NANOADAM implicitly performs weight regularization.

We evaluate NANOADAM across a range of NLP and vision tasks, demonstrating superior memory efficiency and generalization compared to baselines such as MicroAdam, AdamW-8bit, and GaLore. Notably, the efficiency benefits become more pronounced at larger scales. Furthermore, NANOADAM significantly reduces performance degradation on previously learned tasks during continual learning, effectively alleviating catastrophic forgetting.

**Related work**   Fully finetuning a pretrained LLM is known to be resource intensive, prompting substantial work into parameter-efficient-finetuning (PEFT) [12, 34]. Well-known method such as Low Rank Adaptation (LoRA) and its variants [14, 38, 40, 39, 35] update only a set of trainable low-rank matrices while keeping the original model parameters frozen. After finetuning, these low-rank matrices are merged with the original parameters, thereby preserving the inference efficiency. Nonetheless, such low-rank methods can constrain the model's expressiveness [2], limiting its ability to capture complex patterns in new tasks. Furthermore, the introduction of auxiliary parameters leads to an increase in model size during the finetuning process [27].

Several approaches have explored the development of memory-efficient optimizers, motivated by the substantial memory overhead of standard methods like Adam, which require storing multiple optimizer states. Quantization-based techniques, such as Adam-8b [7], reduce the memory footprint by representing optimizer states in lower precision. GaLore [44], on the other hand, maintains full parameter training while enhancing memory efficiency through low-rank gradient factorization. However, the applicability of this type of method is restricted to layers that satisfy the reversibility property, limiting its effectiveness in models lacking this characteristic. Furthermore, the need to perform singular value decomposition (SVD) [43] introduces non-trivial computational overhead. More recently, Low-Dimensional Adam (LDAdam) [30] has been proposed, incorporating low-rank compression for both gradients and optimizer states. To address projection-induced errors, an error feedback mechanism is employed; however, it still necessitates memory to store accumulated errors.

Another line of work focuses on optimizers that update a small subset of trainable parameters without modifying the model architecture or the overall training procedure. For example, SensZOQ [10] finetunes a small, static subset of sensitive parameters, identified by selecting those with the largest squared gradient magnitudes. This approach requires computing the full gradient to determine parameter importance, after which a static mask is applied for zeroth-order finetuning. MicroAdam [24] improves efficiency by dynamically selecting parameters with the top-k gradient values at each optimization step, combined with an error feedback mechanism for correction. However, it still requires maintaining a history of selected gradients and quantized errors, potentially offsetting its memory advantages. Similarly, Dynamic Subset Tuning (DST) [31] updates parameters that exhibit the largest

distance from their pretrained values. Nevertheless, DST must first compute full optimizer updates before selecting the top-k parameters, leading to additional computational overhead. BlockLLM [27] observes that parameters with smaller weight magnitudes tend to be updated more frequently. Nevertheless, it still relies on gradient-based criteria to determine parameter importance. While prior work has largely focused on selecting parameters based on large gradients, one notable exception is [19], which proposes updating small unimportant weights. Their method uses a static mask defined at initialization, without considering dynamic adaptation throughout training as NANOADAM (Further discussion is provided in Appendix). Another related work [29] reduces the number of trainable parameters by randomly selecting a small subset of weights, without leveraging structural signals such as gradients or magnitudes.

## 2 Small weights can matter

Which subset of weights should we update during sparse finetuning? Two main selection principles have been proposed: a) weights with large gradients to approximate full finetuning dynamics [24]; b) weights that are unimportant for the pretrained model to maintain parameters that are important for generic representations, as has been argued in the context of LLM finetuning [19]. Conceptually, both approaches can have pitfalls. a) Full finetuning (and its proxy) could lead to catastrophic forgetting by adapting parameters that capture relevant concepts for both the pretraining and finetuning task. b) Meanwhile, weights that are irrelevant for the pretraining task might also be unimportant for the new task and therefore not contribute meaningfully to learning.

Our observation that large magnitude gradients are associated with small magnitude weights partially reconciles both views. Both subset selection criteria work because they work in tandem. Yet, they still select quite distinct parameter sets. In the following, we argue in favor of updating the smallest weights dynamically during finetuning, as it maintains more relevant information about the pretraining task, enables learning with larger learning rates and thus boosts generalization, and has algorithmic advantages for saving memory. This insight lays the foundation for our proposed dynamic sparse finetuning method, NANOADAM. To provide a theoretical motivation and gain deeper insights into the catastrophic forgetting aspect, we study a two-layer teacher–student network.

### 2.1 Relationship between gradients and weights

We start by investigating the relationship between weights and gradients in finetuning scenarios. Specifically, we fully finetune the BERT-base model on the CoLA dataset from the GLUE benchmark and track the evolution of this relationship for each parameter throughout training. We also conduct a similar experiment in the vision domain, where a ViT-Large model pretrained on ImageNet is finetuned on the CIFAR-10 dataset. To better illustrate the distinction between finetuning and training from scratch, we repeat the vision experiment using the same model architecture but initialized randomly. Details and additional visualizations for these experiments are provided in Appendix B.

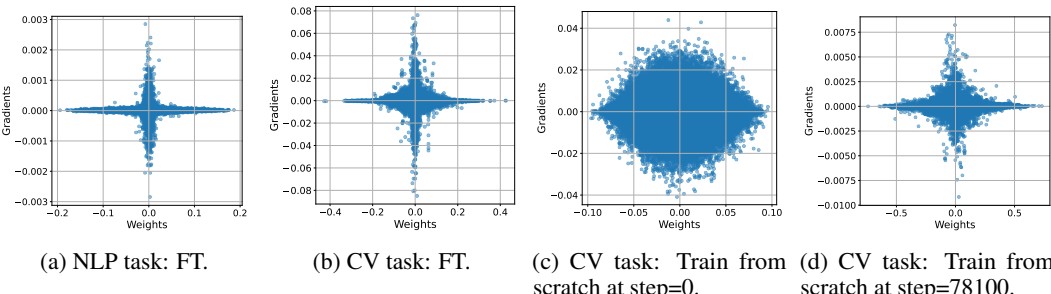

| (a) NLP task: FT. | (b) CV task: FT. | (c) CV task: Train from scratch at step=0. | (d) CV task: Train from scratch at step=78100. |

Figure 1: The relationship between gradients and weights during FT and training from scratch. The x-axis represents the magnitude of the weights, while the y-axis represents the magnitude of the gradients. From left to right, the subfigures correspond to the FT NLP task, FT CV task, training CV task from scratch at early step and training CV task from scratch at later step.

As illustrated in Figure 1, parameters with large gradient magnitudes tend to correspond to small-magnitude weights, with the notable exception of the final classification layer (see Appendix B). This

distinct, hyperbolic relationship is consistently observed across NLP and vision finetuning tasks, indicating that small-magnitude parameters are more actively involved in FT. These findings align with observations of [27], where LLMs were found to update small-magnitude parameters more frequently during adaptation. However, the underlying causes or implications of this phenomenon were neither explored nor exploited. We hypothesize that the strong hyperbolic correlation between gradient and weight may arise from one or more of the following factors: (1) knowledge transferability; (2) overparameterization.

**Knowledge transferability** As shown in Figures 1(b) and (c), finetuning a pretrained ViT-Large model reveals a stronger hyperbolic relationship between gradients and weight magnitudes compared to training from scratch. This likely arises because large weights encode important features learned during pretraining and are therefore less plastic (i.e. are less prone or able to change) —particularly when the finetuning task is similar. Instead, smaller weights adapt more readily to task-specific features. In contrast, training from scratch begins with randomly initialized weights, which lack meaningful structure. Consequently, gradients are more evenly distributed across parameters, resulting in a more elliptical gradient-weight pattern (Figure 1(c)). As training progresses, this distribution gradually shifts toward the hyperbolic form observed in the finetuning regime (Figure 1(d)), reflecting a transition from general feature acquisition to more focused adaptation.

**Overparameterization** Large models are often highly overparameterized, allowing them to adapt to new tasks without significantly modifying their pretrained large weights. To investigate this, we introduce a metric in the gradient–weight space to quantify the degree of hyperbolic correlation. Specifically, we identify the top-$k$ parameters with the largest absolute gradients and compute the *median* of their absolute weights, denoted $w_m$. We take the bottom-$k$ parameters with the smallest absolute gradients and compute the *maximum* of their absolute weights, denoted $w_M$. The ratio

$$r = \frac{\text{median}(|w_i|, \ i \in I)}{\max(|w_i|, \ i \in J)}, \quad I = \text{argmax}_k(|\mathbf{g}|), \quad J = \text{argmin}_k(|\mathbf{g}|) \tag{1}$$

captures the strength of hyperbolic association: a smaller $r$ implies that large gradient parameters tend to have smaller magnitudes, indicating a stronger hyperbolic trend. The use of the median for $w_m$ ensures robustness to outliers.

Table 1: Ratio $r$ for QKV weight matrices at selected layers during early finetuning. Lower $r$ indicates a stronger hyperbolic correlation between gradients and weights.

| Model | Layer 0 | Layer 3 | Layer 6 | Layer 9 | Layer 11 | Layer 17 | Layer 23 |
|---|---|---|---|---|---|---|---|
| ViT-Tiny | 0.200 | 0.040 | 0.120 | 0.070 | 0.200 | – | – |
| ViT-Large | 0.009 | 0.020 | 0.020 | 0.019 | 0.019 | 0.020 | 0.070 |

We evaluate $r$ under two model sizes to study the effect of overparameterization: ViT-Tiny and ViT-Large, both pretrained on ImageNet and finetuned on CIFAR-10 using identical training settings. We compute the ratio using the top 0.01% and bottom 80% of parameters, and report results for QKV weight matrices across layers in Table 1. Visualizations are provided in Appendix D.

As shown, ViT-Large consistently exhibits lower $r$ values, supporting the hypothesis that overparameterized models develop a stronger hyperbolic gradient–weight structure. This insight motivates the central question of our work: *Can parameter magnitude—rather than gradient magnitude—serve as a more effective criterion for selecting which subset of parameters to update during finetuning?* We explore this both theoretically and empirically next.

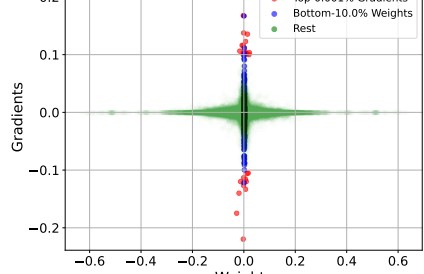

Figure 2: Overlap between small weights and large gradients.

## 2.2 Small weights and large gradients select distinct parameter subsets

To further investigate the relationship between weights and gradients, we examine two parameter subsets: the top 0.001% with the largest absolute gradients and the bottom 10% with the smallest weight magnitudes,

based on Figure 1(b). This analysis aims to determine whether small weights tend to coincide with large gradients. The overlap is visualized in Figure 2. Interestingly, even the bottom 10% of small weights fail to fully cover the top 0.001% of large gradients. This suggests that small weights and large gradients are not simply two sides of the same coin. Accordingly, also parameters receive large gradients that are deemed important for the pretraining task and their adaptation could lead to forgetting transferable concepts.

In contrast, we argue that focusing on small weights during finetuning has several advantages: (1) Modern neural networks are often heavily overparameterized, making it sufficient to learn downstream tasks by updating only small-weight parameters; (2) While not perfectly aligned, small weights have a non-trivial chance of intersecting with large gradients. When combined with a dynamic masking schedule, this ensures broader coverage over time; (3) Large weights likely encode essential features from pretraining, and modifying them risks disrupting previously learned knowledge. Focusing updates on small weights introduces minimal interference with this structure. As shown in our experiments later, updating small weights achieves superior generalization and results in smaller overall parameter shifts. This suggests that finetuning small weights follows a distinct and efficient training dynamic, rather than approximating the path of full or large gradient-driven updates.

### 2.3 Nano gradient flow: From feature learning to finetuning

We provide a theoretical motivation and visual illustration explaining why updating small weights could be more beneficial than updating large gradients.

**Setup** Consider a student-teacher setup based on two-layer neural networks [5]. Specifically, let $f : \mathbb{R}^n \times \mathbb{R}^{n \times d} \to \mathbb{R}$ denote a two-layer neural network with parameters $(a, W)$ and input $x$, $f(a, W|x) := \sum_{i=1}^n a_i \sigma(w_i x)$ , where $\sigma(\cdot) = \max\{0, \cdot\}$ is the ReLU activation. All networks in this section follow this form. We first pretrain a student network $f_{\text{pre}}$ in the feature learning regime using the mean squared error (MSE) loss and gradient descent for $T$ steps. The inputs $\{x_j\}_{j=1}^k$ is sampled i.i.d. from a multivariate Gaussian $N(0, I_d)$, where $I_d$ is the $d-$dimensional identity matrix. The targets are generated by a teacher network $f_{\text{teacher}}$. To simulate finetuning, we perturb the teacher network $f_{\text{teacher}}$ with an additional neuron, yielding $f_{\text{finetune}} = f_{\text{teacher}} + f_{\text{extra}}$. During finetuning, we compare two strategies: updating parameters in each layer with either the largest gradients or the smallest magnitudes. A case study involving multiple additional neurons is included in Appendix E, along with the corresponding hyperparameter settings for these experiments.

**Theoretical motivation** One of our goals is to preserve the original representation during finetuning, i.e., to mitigate catastrophic forgetting. In a two-layer neural network, this corresponds to retaining the largest neurons—those with the largest activations. These can be ordered by their effective magnitude, $|a_i|\|w_i\|$. We show that training only the small-magnitude weights preserves the representation. We formalize this idea in the following definition.

**Definition 2.1** *A finetuned network $f$ is $k$-neuron representation preserving iff the largest neurons corresponding to Top-$k(j \in [n] : |a_j|\|w_j\|)$ remain unchanged compared to the pre-training task.*

We assume pretraining occurs in the feature learning regime [5]. Due to the implicit sparsity bias in over-

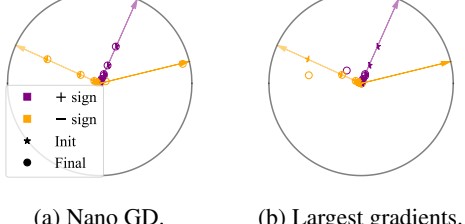

(a) Nano GD.  (b) Largest gradients.

Figure 3: Nano gradient descent provably prevents catastrophic forgetting. (a) Nano gradient descent keeps the original representation while learning the extra neuron. (b) The largest gradients can correspond to weights with large magnitudes leading to unlearning of the original representation and the inability of learning the new representation.

parameterized training, there often exists at least one sufficiently small neuron that has minimal impact on the pretrained representation. We analyze the effect of updating only the smallest neuron:

**Theorem 2.2** *Assume a model $f(x)$ consisting of $n$ neurons learns the teacher $f_{teacher}(x)$ corresponding to a pre-training task so that $f(x) = f_{teacher}(x)$ for all $x \in \mathbb{R}^d$. Furthermore, let $f(x)$ consist of at least two neurons $i, r \in [n]$ such that $\max\{|a_i|^2, |a_r|^2\} \leq \epsilon$ for an $\epsilon > 0$ and $sign(a_i) \neq sign(a_r)$. Let a new task be defined based on labels $f_{finetune} := f_{teacher} + f_{extra}$ with an*

*extra neuron $f_{extra} = \tilde{a}\sigma(\tilde{w}\cdot)$. Let only the neuron $j$ of $f$ be trainable to finetune $f(x)$ to the new task, where $j = argmin\{|a_i|\|w_i\| : sign(a_i) = sign(\tilde{a})\}$. Then, the gradient flow with respect to finetuning time $t$ of the neuron $j$, which is parameterized as $v_{j,t} = |a_{j,t}|w_{j,t}$ and initialized at the pre-trained values $v_{j,0} = |a_j|w_j$, converges to a value $v_\infty$ so that $\|v_\infty - v\|_{L_2} < C\epsilon$, where $v$ is the target $v = |\tilde{a}|\tilde{w}$ and $C > 0$ a data dependent constant.*

*Proof.* The proof follows from using Theorem 6.4 [23] for learning with one neuron a one neuron target and controlling the perturbation incurred by the difference $f_{\text{teacher}} - \sigma(v_0)$ i.e. the small $\epsilon$ trainable neuron $v$. (See Theorem E.1) □

Theorem 2.2 indicates that updating the smallest neuron is sufficient for learning new representations without disrupting the pretrained structure. A key mechanism underlying the nano gradient flow (learning rate $\eta \to 0$, see Appendix E) is: *In the feature learning regime [9, 1], gradient flow satisfies:* $|a_{i,t}|^2 = \|w_{i,t}\|^2$ *for all $i \in [n]$ and $t \geq 0$.* This observation implies that selecting the smallest weights in each layer corresponds to training the smallest neuron. This allows nano gradient flow to learn new task-specific information while preserving the original representation, thereby reducing catastrophic forgetting. In contrast, selecting large-gradient parameters updates large, pretrained neurons and risks overwriting important features. This highlights a more general principle by training the smallest weights: *Nano gradient flow or* NANOADAM *learns a compact task-specific representation, while (partially) preserving the pretrained representation.*

**Catastrophic forgetting** We construct a teacher network with two neurons and pretrain a two-layer student network in the feature learning regime [5]. To simulate a finetuning scenario, we define a new task by adding a randomly initialized neuron and generating labels accordingly. See Appendix E for full details. We select the parameters with either the largest gradients or smallest magnitudes.

In Figure 3, the teacher neurons are represented as arrows $|a|w$. pretrained or finetuned neurons are visualized as points. The color indicates the sign of $a$. According to Figure 3a, selecting the smallest weights allows the network to learn the new representation while preserving the pretrained one. In contrast, as seen in Figure 3b, large gradients can interfere with large weights and degrade the original knowledge. Table 8 in Appendix E confirms that small-weight updates result in better generalization and smaller $\ell_2$ shifts, indicating greater representation retention. The measure is motivated by Lemma E.2 in the appendix.

## 3 NANOADAM

Motivated by the empirical findings and theoretical insights discussed above, we introduce NANOADAM, an optimizer that finetunes a subset of parameters based solely on their weight magnitudes, as outlined in Algorithm 1. To further enhance efficiency, we incorporate a density scheduler that dynamically adjusts the fraction of updated parameters during training. A detailed memory analysis is provided in Appendix G.

We adopt standard Adam-like notation: let $m_t$ and $v_t$ denote the first- and second-order momentum estimates of the gradients at step $t$, with momentum coefficients $\beta_1$ and $\beta_2$, and a small constant $\epsilon$ for numerical stability. Let $f$ be the loss function, $\theta_t$ the model parameters at step $t$, and $\eta$ the learning rate. The full gradient is denoted by $\nabla_\theta f$. A mask $I$ indexes the selected subset of parameters to update. The density of this subset is denoted by $k$, while $m$ and $d$ represent the update intervals for the mask and density, respectively. Finally, $T$ denotes the total number of optimization steps.

### 3.1 Algorithm details

The core idea behind NANOADAM is to determine a mask for selected parameters solely

---

**Algorithm 1** NANOADAM

**Require:** initial density $k_0$, mask interval $m$, density interval $d$, total steps T, $\beta_1$, $\beta_2$
1: $m_0, v_0, I, k \leftarrow 0, 0, 0, k_0$
2: **for** $t = 0$ to T **do**
3: $\quad$ flag$_k \leftarrow False$
4: $\quad$ **if** $t\%d == 0$ **then**
5: $\quad\quad k \leftarrow$ density schedule$(k, t, T)$
6: $\quad\quad$ flag$_k \leftarrow True$
7: $\quad$ **end if**
8: $\quad$ **if** $t\%m == 0$ or flag$_k == True$ **then**
9: $\quad\quad I \leftarrow$ Bottom$_k(|\theta_t|)$
10: $\quad$ **end if**
10: $\quad g_t \leftarrow \nabla_\theta f(\theta_t)[I]$
10: $\quad m_t \leftarrow$ momentum update$(m_{t-1}, g_t, \beta_1)$
10: $\quad v_t \leftarrow$ momentum update$(v_{t-1}, g_t, \beta_2)$
10: $\quad \theta_{t+1} \leftarrow \theta_t - \eta_t \frac{m_t}{\sqrt{v_t}+\epsilon}$
11: **end for**

---

based on their absolute magnitudes, selecting the subset with the smallest values. Given that small-magnitude parameters tend to remain small throughout optimization, it is unnecessary to update the mask at every step. Instead, we introduce a mask interval $m$, such that the mask $I$ is only updated once every $m$ steps. This design provides two key benefits. First, it improves computational efficiency by reducing the overhead associated with frequent mask updates. Second, it enables the optimizer to preserve the momentum-like dynamics of Adam, while maintaining the first- and second-order momentum only for the selected subset, thereby reducing memory consumption. To further enhance memory efficiency, we incorporate a density scheduler, akin to a learning rate scheduler, that dynamically adjusts the density $k$ throughout training. By default, we employ a linear decay schedule, though this mechanism can be disabled by setting the density update interval $d$ greater than the total number of training steps $T$.

Importantly, NANOADAM does not incorporate any feedback mechanism to compensate for the error introduced by gradient sparsification. This exclusion is a deliberate design choice based on three key considerations: (1) Error feedback mechanisms introduce additional memory overhead for storing residuals and computational overhead for accumulation and reinsertion—contrary to the goal of optimizing efficiency; (2) NANOADAM does not aim to approximate the trajectory of full-gradient updates, but instead pursues a distinct and efficient optimization path; and (3) Empirically, we find that incorporating error feedback offers no performance benefit and can even degrade generalization.

## 3.2 Ablation study

To validate the effectiveness of our method, we conduct ablation studies comparing various masking strategies: small weights vs. large or random weights, and small weights vs. large gradients. An additional study comparing static and dynamic masking strategies is included in Appendix F.3.

**Small vs. large vs. random weights** We begin by evaluating the impact of different weight-based masking strategies in an LLM finetuning setup. Specifically, we finetune the BERT-base model on the SST-2 task from the GLUE benchmark using NANOADAM under three masking strategies: (1) small weights, (2) large weights, and (3) random weights. Importantly, each configuration uses the same gradient density $k$. (See Appendix F.1 for details.)

Figure 4a shows the generalization results, with training loss dynamics available in Appendix F.1. Finetuning small-magnitude weights consistently yields the lowest training loss and highest evaluation accuracy. In contrast, updating large weights results in the worst performance—even random masking performs better. This could suggest that most large weights, which likely encode critical pretrained features, are less adaptable. In comparison, small weights exhibit greater plasticity, enabling efficient adaptation while preserving core model capabilities. This also relates to catastrophic forgetting: updating large weights risks overwriting pretrained knowledge, whereas small weights provide a safer avenue for learning.

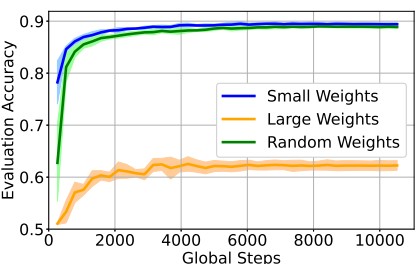

(a) Small vs. large vs. random weights.

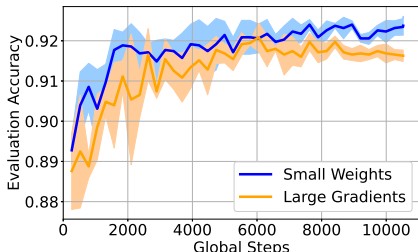

(b) Small weights vs. large gradients.

Figure 4: Generalization performance of different masking strategies in NANOADAM using the same gradient density. (a) Small vs. large vs. random weights. (b) Small weights vs. large gradients. Small-weight masking achieves the best generalization performance.

**Small weights vs. large gradients** We further investigate whether small weights offer a more effective selection criterion than large gradients. Under the same experimental setup, we finetune the BERT-base model on SST-2 using a dynamic masking interval of $m = 131$ steps. Two selection strategies are compared: (1) parameters with large gradients and (2) parameters with small weights. Given that small-weight parameters typically require higher learning rates for effective updates, we perform hyperparameter tuning for both strategies. The optimal learning rate is $1 \times 10^{-3}$ for small weights and $3 \times 10^{-4}$ for large gradients. Notably, applying $1 \times 10^{-3}$ to the large-gradient strategy causes divergence. Figure 4b shows the evaluation results; training losses are included in Appendix F.2. Results show that updating small weights leads to faster convergence and better

generalization, supporting the view that they are more plastic and better suited for finetuning. A corresponding study in the vision domain, provided in Appendix F.2, confirms these findings.

## 4 Experiments

We evaluate the effectiveness of NANOADAM on both NLP and CV finetuning tasks. Our experiments compare against several baselines, including AdamW [22], AdamW-8bit [7], GaLore [44], and MicroAdam [24]. For NLP, we evaluate three language models of varying scale: BERT-Base (110M parameters), BERT-Large (335M) [8], and OPT-1.3B [42]. These models are finetuned across multiple tasks from the GLUE benchmark. For CV, we examine two aspects: catastrophic forgetting and parameter shift. Specifically, we finetune a ViT-Large [36], ResNet101, and ResNet18 [13], all pretrained on ImageNet [6]. Each model is first finetuned on CIFAR-10 [18], followed by continued finetuning on the Flowers dataset [25]. Complete training configurations, hyperparameter details, and additional results—including the learning rate study—are provided in Appendix H and Appendix I.

**Finetuning on NLP tasks** We first evaluate the effectiveness of NANOADAM on NLP finetuning tasks from the GLUE benchmark. Our experiments use Transformer models from the HuggingFace library, including BERT and OPT-1.3B. For performance evaluation, we use standard metrics: matched accuracy for MNLI, Matthew's correlation for CoLA, Pearson correlation for STS-B, and classification accuracy for the remaining tasks. All optimizers are evaluated under consistent training conditions, with the exception that the learning rate is individually tuned. Experiments are conducted on a compute node equipped with 4×A100 40GB GPUs. Memory usage is reported as the average across all GPUs. The overall performance results are summarized in Table 2 and 3, while details on peak memory usage, training time, and training dynamics are deferred to Appendix H.2.

Table 2: Performance (eval metric) on GLUE dataset.

| Model | Method | COLA | SST2 | MRPC | STSB | QQP | MNLI | QNLI | AVG. |
|---|---|---|---|---|---|---|---|---|---|
| BERT-BASE | Microadam | 60.26 | 92.89 | 83.82 | 88.72 | 90.63 | 84.04 | 91.18 | 84.50 |
| | NANOADAM | 60.87 | 93.46 | 88.48 | 89.98 | 90.67 | 84.30 | 91.76 | **85.65** |
| | Galore | 57.90 | 92.20 | 85.54 | 89.90 | 89.91 | 82.81 | 90.87 | 84.16 |
| | AdamW-8b | 60.41 | 93.01 | 87.26 | 89.68 | 90.70 | 84.16 | 91.40 | 85.23 |
| | AdamW | 59.65 | 93.23 | 87.01 | 87.90 | 89.66 | 83.29 | 91.31 | 84.58 |
| BERT-Large | Microadam | 62.55 | 94.04 | 89.22 | 89.68 | 90.45 | 85.67 | 92.04 | 86.24 |
| | NANOADAM | 66.85 | 94.61 | 90.20 | 90.86 | 91.03 | 86.40 | 92.44 | **87.48** |
| | Galore | 61.46 | 94.27 | 87.01 | 89.08 | 89.73 | 84.95 | 91.58 | 85.44 |
| | AdamW-8b | 63.95 | 94.38 | 88.97 | 90.04 | 91.35 | 86.31 | 92.37 | 86.77 |
| | AdamW | 61.53 | 94.15 | 86.03 | 89.74 | 90.05 | 86.09 | 92.18 | 85.68 |
| OPT-1.3B | Microadam | 66.80 | 95.99 | 88.24 | 89.66 | 91.51 | 87.94 | 92.73 | 87.55 |
| | NANOADAM | 67.69 | 96.45 | 87.99 | 91.00 | 91.33 | 88.24 | 92.75 | **87.92** |
| | Galore | 65.88 | 96.10 | 86.03 | 90.86 | 90.80 | 87.89 | 92.72 | 87.18 |
| | AdamW-8b | 66.36 | 95.87 | 86.28 | 90.36 | 91.57 | 87.20 | 92.79 | 87.20 |
| | AdamW | 66.50 | 95.64 | 85.29 | 90.28 | 91.34 | 87.86 | 92.93 | 87.12 |

The results show that NANOADAM achieves lower memory usage than other memory-efficient optimizers, including AdamW-8bit, GaLore, and MicroAdam, while also delivering superior generalization performance. Notably, the memory savings scale with model size, in line with our theoretical analysis. Additionally, while methods like MicroAdam and GaLore suffer from significantly higher training time on larger models, NANOADAM maintains comparable runtime efficiency to well-optimized baselines such as AdamW and AdamW-8bit.

**Catastrophic forgetting on CV tasks** We evaluate the catastrophic forgetting behavior of AdamW, MicroAdam, and NANOADAM across several vision models, including ViT-Large, ResNet101, and ResNet18, in a continual learning setting. Each model is first finetuned on CIFAR-10 (Task 1) for a fixed number of epochs, followed by continued finetuning on Flowers102 (Task 2). To assess forgetting, we measure generalization performance on: (1) CIFAR-10 after Task 1, (2) Flowers102

Table 3: Average memory usage (GB) on GLUE dataset.

| Model | MicroAdam | NanoAdam | GaLore | AdamW-8b | AdamW |
|---|---|---|---|---|---|
| BERT-Base | 3.71 | **3.58** | 4.04 | 3.72 | 3.94 |
| BERT-Large | 5.54 | **5.18** | 5.91 | 5.64 | 6.48 |
| OPT-1.3B | 13.18 | **11.60** | 14.16 | 13.08 | 18.16 |

after Task 2, and (3) CIFAR-10 again after Task 2. Forgetting is quantified as the drop in CIFAR-10 accuracy before and after Task 2. Note that CIFAR-10 and Flowers102 differ in the number of classes. Thus, to evaluate CIFAR-10 performance after Task 2, we reload the original classification head trained on Task 1. This isolates representational drift and allows us to assess the extent to which pretrained features are preserved. The resulting generalization performances are summarized in Table 4, with experimental details and visualisations of training dynamics provided in Appendix I. As shown, while all methods perform well on CIFAR-10 after the initial finetuning stage, most suffer substantial degradation after Task 2—indicating significant catastrophic forgetting. In contrast, for ResNet101, NANOADAM preserves high accuracy on CIFAR-10 (83.77%) and also adapts well to Flowers102 (81.52%), outperforming AdamW on both tasks (29.59% and 52.97%, respectively). These results suggest that NANOADAM achieves a more favorable trade-off between knowledge retention and task adaptation.

Table 4: Evaluation accuracy (%) across tasks for catastrophic forgetting.

| Model | Method | CIFAR-10 (Task 1) | Flowers102 (Task 2) | CIFAR-10 (after Task 2) | Avg. | Forgetting |
|---|---|---|---|---|---|---|
| | MicroAdam | 99.12 | 88.23 | 98.36 | 95.23 | 0.76 |
| ViT-Large | NANOADAM | 99.37 | 98.13 | 99.35 | **98.95** | **0.02** |
| | AdamW | 99.3 | 92.51 | 98.61 | 96.81 | 0.69 |
| | MicroAdam | 95.47 | 14.89 | 12.49 | 40.95 | 82.98 |
| ResNet101 | NANOADAM | 96.32 | 81.52 | 83.77 | **87.20** | **12.55** |
| | AdamW | 97.72 | 52.97 | 29.59 | 60.09 | 68.14 |
| | MicroAdam | 93.12 | 22.53 | 25.36 | 47.00 | 67.76 |
| ResNet18 | NANOADAM | 92.52 | 70.67 | 64.17 | **75.79** | **28.36** |
| | AdamW | 95.63 | 56.58 | 27.59 | 59.93 | 68.05 |

**Parameter shift analysis**  We further analyze the extent of parameter changes during finetuning under different optimizers. Specifically, we compute the $\ell_2$ distance between the pretrained ViT-Large parameters and those obtained after continual finetuning on CIFAR10 and Flowers, with MicroAdam, AdamW, and NANOADAM. The classification head is excluded from this analysis to isolate changes in the backbone. As in our toy example, Table 5 summarizes the average $\ell_2$ distance in parameter change alongside the average evaluation accuracy. Despite using the largest learning rate, NANOADAM induces the smallest parameter shift and achieves the best generalisation. More details are provided by visualizations in Appendix I.3.

Table 5: Averaged evaluation accuracy and parameter change in $\ell_2$ distance, alongside learning rate.

| Algorithm | LR (task1) | LR (task2) | AVG. Acc | $\ell_2$ Distance |
|---|---|---|---|---|
| AdamW | 1e−4 | 1e−4 | 96.81% | 0.83 |
| MicroAdam | 1e−4 | 1e−3 | 95.23% | 0.75 |
| NanoAdam | 1e−3 | 2e−3 | **98.95%** | **0.68** |

## 5  Conclusions

We introduce NANOADAM, a memory- and compute-efficient optimizer for finetuning large models. Motivated by a consistent hyperbolic correlation between gradients and small weights observed

during finetuning, we propose to dynamically update parameters with small magnitudes instead of large gradients. Although this relationship is less evident when training from scratch, it proves highly effective in finetuning scenarios, where avoiding forgetting relevant concepts is paramount. Unlike prior methods, NANOADAM selects parameters without relying on gradient information, leading to improved generalization, less catastrophic forgetting, and reduced parameter drift. Experiments on both NLP and vision tasks show that NANOADAM matches or outperforms existing methods, offering a new perspective on the role of small weights in efficient finetuning.

## 6 Limitations and broader implications

Our proposed method NanoAdam introduces minimal computational overhead. Specifically, for each weight matrix in each layer, we first flatten the matrix and divide it into subgroups (chunks), then apply bottom-k selection within each subgroup. This process is applied uniformly across both convolutional and MLP layers. The main computational cost arises from the bottom-k operation, which has a time complexity of $O(k \log k)$.

Thanks to its layer-wise and parameter-wise design, the method is naturally scalable to larger models and remains compatible with modern hardware acceleration and parallel training frameworks. It avoids conflicts with model parallelism and pipeline layers, making it practical for contemporary large-scale architectures.

However, the method has several limitations. Its effectiveness relies heavily on knowledge transferability and overparameterization. When the pretraining and finetuning tasks are well-aligned, the method helps avoid catastrophic forgetting and effectively leverages the plasticity of small weights to adapt to new tasks. In contrast, when there is limited similarity between tasks, the method may underperform compared to full-update optimizers like Adam. Moreover, the method benefits significantly from model overparameterization. As demonstrated in our experiments on vision tasks, scaling from a smaller model (e.g., ResNet-18) to a larger one (e.g., ViT-Large) results in improved overall performance and reduced forgetting. This suggests that NanoAdam the method is particularly well-suited for large, overparameterized models.

## 7 Acknowledgements

The authors gratefully acknowledge the Gauss Centre for Supercomputing e.V. (www.gauss-centre.eu) for funding this project by providing computing time on the GCS Supercomputer JUWELS [15] at Jülich Supercomputing Centre (JSC). We are also grateful for funding from the European Research Council (ERC) under the Horizon Europe Framework Programme (HORIZON) for proposal number 101116395 SPARSE-ML.

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

# A  Contribution related to literature

Our work is different from [19] from the following aspects:

1. While [19] hypothesizes that updating only small-magnitude weights can be effective, we provide an explanation: large gradients are usually associated with small weights during finetuning, making them more adaptable. We show that the gradient-weight correlation is much stronger during fine-tuning than training from scratch, explaining why the strategy is less effective in the latter case.

2. We attribute this pattern to knowledge transferability and overparameterization, offering a principled understanding of when small-weight finetuning is most effective.

3. We further show that updating small weights helps mitigate catastrophic forgetting, supported by both theoretical and empirical evidence.

4. Our method is novel in that: (a) mask selection and (b) sparsity are dynamic, and (c) selection is done per layer rather than globally. Unlike [18], which uses a fixed global mask, our dynamic approach improves efficiency and performance while reducing memory usage. Dynamic masking specifically improves the implicit weight regularization by discouraging over-reliance on a fixed subset of parameters; and mitigates catastrophic forgetting by reducing parameter shift.

5. Lastly, unlike [18] which focuses only on NLP, we evaluate our method on CV tasks as well, demonstrating broader applicability.

# B  Details on studying the relationship between gradients and weights in Section 2.1

The training details used are summarised in Table 6. In Figure 5, we illustrate how the relationship between weights and gradients evolves for three representative components: the positional embedding weights, the value weight matrix in layer 6, and the weight matrix in the final classification layer, corresponding to progressively deeper layers in the network.

As shown in the figure, parameters with large gradients typically correspond to those with small weight magnitudes (except in the case of the final classification layer as shown in the appendix). There are several possible explanations for the distinct behaviour observed in the final classifier layer. First, the classifier is newly initialised from scratch, rather than being inherited from pretrained weights. Second, it needs to adapt to the task-specific label space. Consequently, in our algorithm, we exclude the final layer, along with normalisation layers, from selective updates. In contrast, for the other layers, the strong correlation between gradient magnitude and weight magnitude remains significant. This finding also supports the observation made in [27], where models were found to frequently update parameters with smaller weight magnitudes. However, in their work, this phenomenon was not further explored; instead, they adhered to prior practice by using gradients as the primary criterion for parameter importance.

Table 6: Hyperparameters for studying the gradient-weight relationships.

| Setting | LR | WD | Batch size | Epoch | Label Smooth |
|---|---|---|---|---|---|
| CoLA (BERT-base, FT) | 7e−5 | 0.0 | 32 | 5 | – |
| CIFAR10 (ViT, FT) | 3e−3 | 0.1 | 128 | 300 | 0.1 |
| CIFAR10 (ViT, Scratch) | 3e−3 | 0.1 | 128 | 300 | 0.1 |

# C  Small weights and large updates

To better understand the actual parameter changes during fine-tuning, we conducted an additional experiment: we fine-tuned a BERT-base model on the CoLA task from the GLUE benchmark using AdamW for 5 epochs.

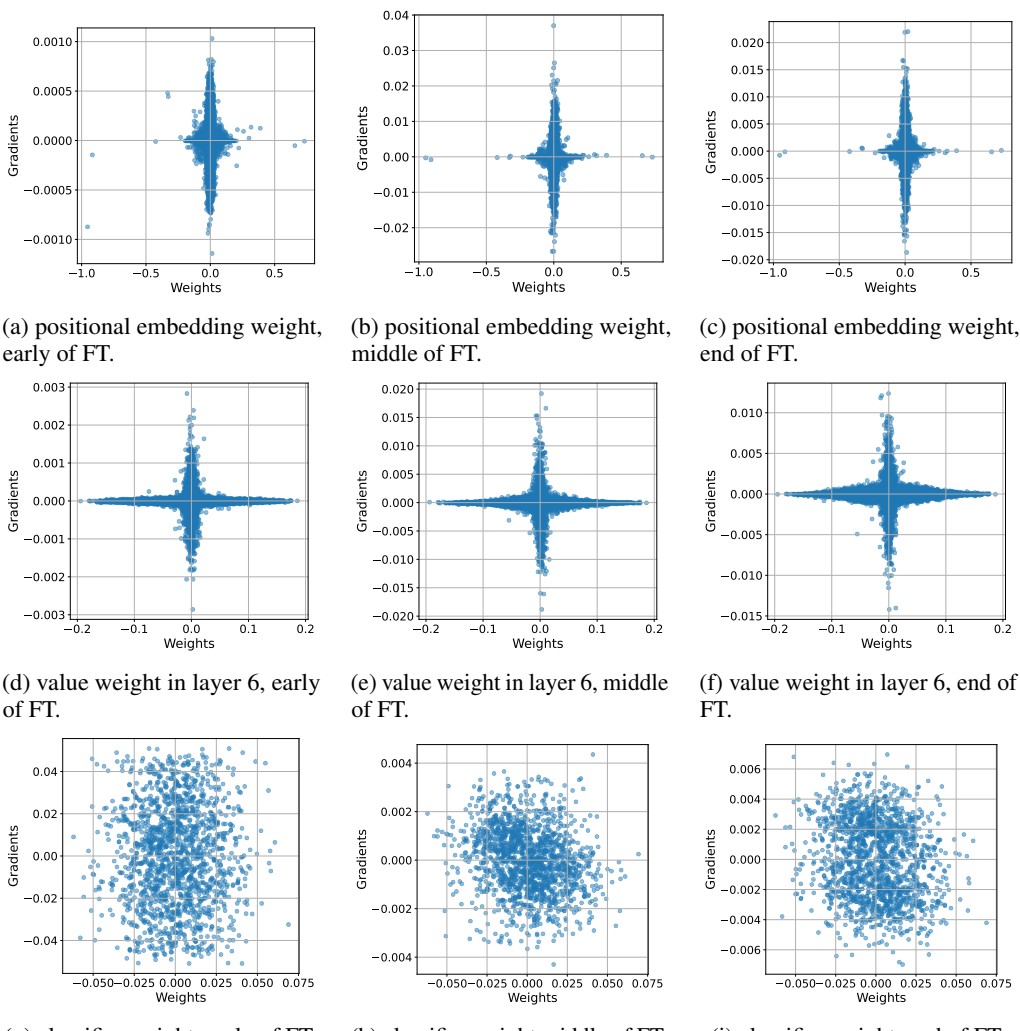

(a) positional embedding weight, early of FT.

(b) positional embedding weight, middle of FT.

(c) positional embedding weight, end of FT.

(d) value weight in layer 6, early of FT.

(e) value weight in layer 6, middle of FT.

(f) value weight in layer 6, end of FT.

(g) classifier weight, early of FT.

(h) classifier weight middle of FT.

(i) classifier weight, end of FT.

Figure 5: The dynamic of the relationship between gradients and weights during finetuning Bert-base on COLA. The x-axis represents the magnitude of the weights, while the y-axis represents the magnitude of the gradients. From left to right, the subfigures correspond to the early, middle, and late stages of finetuning. From top to bottom, the subfigures represent progressively deeper layers in the network.

For each layer, we partitioned the parameters into three groups based on their absolute magnitudes in the pretrained model:

1. The bottom 30% (smallest magnitudes);

2. The top 30% (largest magnitudes);

3. The remaining middle 40%.

We then measured the average change in weights within each group, calculated as the difference between the pretrained and fine-tuned weights. This analysis allows us to examine which groups of parameters receive the largest updates during fine-tuning.

The results, presented in the table 7 below, consistently show that across all layers, the smallest-magnitude weights undergo the largest updates, while the largest-magnitude weights change the least. This trend holds regardless of the layer's scale or functional role in the Transformer architecture.

Table 7: Per-layer parameter shift for the 30% smallest, 30% largest, and remaining subsets.

| Layer | 30% Smallest | 30% Largest | Remaining |
|---|---|---|---|
| Layer 0 | 0.000799 | 0.000381 | 0.000393 |
| Layer 1 | 0.000815 | 0.000325 | 0.000363 |
| Layer 2 | 0.000787 | 0.000331 | 0.000344 |
| Layer 3 | 0.000812 | 0.000302 | 0.000353 |
| Layer 4 | 0.000846 | 0.000298 | 0.000354 |
| Layer 5 | 0.000877 | 0.000282 | 0.000351 |
| Layer 6 | 0.000821 | 0.000294 | 0.000342 |
| Layer 7 | 0.000798 | 0.000299 | 0.000341 |
| Layer 8 | 0.000727 | 0.000276 | 0.000321 |
| Layer 9 | 0.000691 | 0.000278 | 0.000308 |
| Layer 10 | 0.000669 | 0.000280 | 0.000302 |
| Layer 11 | 0.000742 | 0.000279 | 0.000324 |

## D  Overparameterization leads to stronger correlation

We provide additional visualisation of gradient-weights distribution in FT ViT-Tiny and ViT-Large models on CIFAR10 in Figure 6.

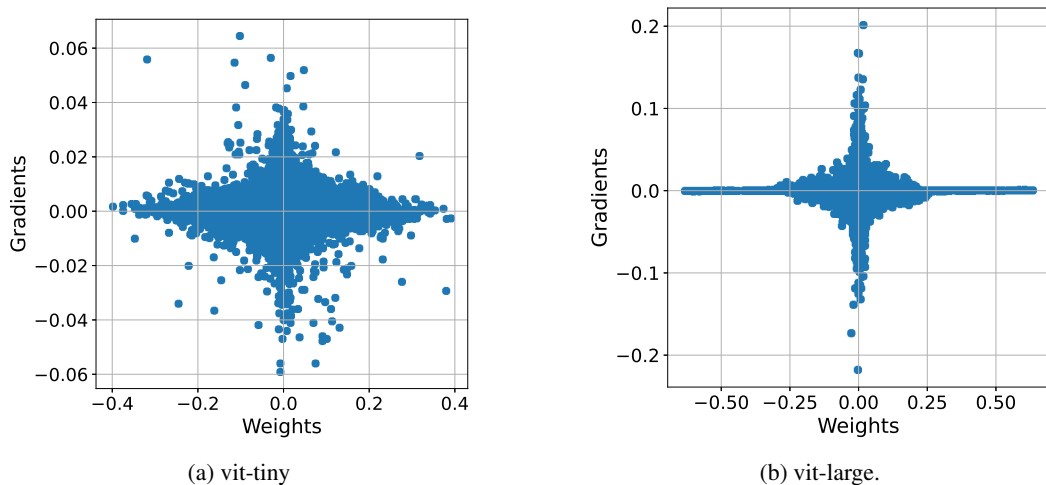

(a) vit-tiny                    (b) vit-large.

Figure 6: Gradients and weights distribution of first layer in FT ViT-Tiny and ViT-Large at early stage of FT. Overparmeterisation leads to more hyperbolic relationship.

# E Additional theoretical statements and motivation

We provide more details regarding the two layers neural network setup. For pretraining, the data is generated from a similar structured teacher network $f_{\text{teacher}}$ with $n = 2$ neurons and input dimension $d = 2$. Furthermore, all neurons are first initialized $a \sim Uni(\{-1, 1\})$ i.e. i.i.d. Rademacher random variables and $w_{i,j} \sim N(0, 1)$ for all $i, j \in [k, d]$ and normalized over the input dimension. We initialize the dense network with $k = 20$ and the so-called COB initialization based on the rich regime in [5]. All weights are initialized with $\mathcal{N}(0, 1/n)$ and we ensure that $a_i = -a_{i+10}$ for $i \in [10]$. We train for $T = 10000$ steps with learning rate $\eta = 2$. This training gives us the pretrained network $f_{\text{pre}}$. For the fine tuning we generate additional neurons in the same way as the teacher neuron. We train for $T = 10000$ steps with learning rate $\eta = 1$. A slightly smaller learning rate has been chosen to ensure convergence for the large gradient setup.

**Gradient flow**   The gradient flow of a two layer neural network for training one neuron is given by

$$\begin{cases} da_t = -\frac{1}{n} \sum_i \sigma\left(w_t^T x_i\right)\left(a_t \sigma\left(w_t^T x_i\right) - y_i\right) dt, & a_0 = a_{\text{init}} \\ dw_t = -\frac{1}{n} \sum_i a_t x_i \mathbb{I}_{w_t^T x_i > 0}\left(a_t \sigma\left(w_t^T x_i\right) - y_i\right) dt, & w_0 = w_{\text{init}}, \end{cases}$$

As highlighted the study of nano gradient flow can be reduced to studying the case of training one neuron where the labels are generated by a teacher neuron $f_{\text{extra}}$. This is under the assumption that the chosen neuron is not contributing to the representation. In our main statement we simplify even further and train one neuron with $a$ frozen as in the next statement Theorem E.1.

**Theorem E.1** *Assume a model $f(x)$ consisting of $n$ neurons learns the teacher $f_{teacher}(x)$ corresponding to a pre-training task so that $f(x) = f_{teacher}(x)$ for all $x \in \mathbb{R}^d$. Furthermore, let $f(x)$ consist of at least two neurons $i, r \in [n]$ such that $\max\{|a_i|^2, |a_r|^2\} \leq \epsilon$ for an $\epsilon > 0$ and $sign(a_i) \neq sign(a_r)$. Let a new task be defined based on labels $f_{finetune} := f_{teacher} + f_{extra}$ with an extra neuron $f_{extra} = \tilde{a}\sigma(\tilde{w}\cdot)$. Let only the neuron $j$ of $f$ be trainable to finetune $f(x)$ to the new task, where $j = argmin\{|a_i|||w_i|| : sign(a_i) = sign(\tilde{a})\}$. Then, the gradient flow with respect to finetuning time $t$ of the neuron $j$, which is parameterized as $v_{j,t} = |a_{j,t}|w_{j,t}$ and initialized at the pre-trained values $v_{j,0} = |a_j|w_j$, converges to a value $v_\infty$ so that $||v_\infty - v||_{L_2} < C\epsilon$, where $v$ is the target $v = |\tilde{a}|\tilde{w}$ and $C > 0$ a data dependent constant.*

Proof. To apply Theorem 6.4 in [23], we need matching signs for the parameters $a$. Otherwise, we have an immediate mismatch between the two single neuron functions. Assuming matching signs, we can absorb $a_{min}$ and $\tilde{a}$ into the activation to simplify the analysis. This reduces the problem to optimizing a single layer neuron $\sigma(v_t\cdot)$ with initialization $|a_j|w_j$ to learn a target vector $v = |\tilde{a}|\tilde{w}$ with some small label perturbation that is equal to $\epsilon$. Without loss of generality we assume the sign of $a$ is positive. Then setting is reduced to training one neuron with gradient flow:

$$dv_t = -\frac{1}{n} \sum_i x_i \mathbb{I}_{v_t x_i \geq 0}\left(\sigma(v_t x_i) - \sigma(v x_i) + B_{\epsilon,i}\right) dt,$$

where $|B_{\epsilon,i}| \leq C_1 \epsilon$ is a small perturbation incurred from the teacher and $C_1 > 0$ is data dependent constant. We can characterize the minumum associated with $v$ using perturbation theory i.e. we can linearize around $v$ and $\epsilon = 0$ the right hand side of the gradient flow equation:

$$-v_0' \frac{1}{n} \sum_i x_i x_i^T \mathbb{I}_{x_i v \geq 0} + C = 0$$

giving us $v_0' = H^{-1} C$, where $H$ is the Hessian or data covariance matrix and $C \in \mathbb{R}^d$ depends on all $B_{\epsilon,i}$ for $i \in [n]$. This leads to a bound for the perturbed equilibrium $v^*$:

$$||v^* - v||_{L_2} \leq ||v'||_{L_2} \epsilon.$$

Denote the process $\hat{v}_t$ as the gradient flow without perturbation. It follows directly from Theorem 6.4 in [23] that $\hat{v}_t \to v$. It remains to be shown that $\hat{v}_t$ is close to $v_t$ during the gradient flow. At initialization, we have $v_0 = \hat{v}_0$. We can bound the evolution with $z_t := v_t - \hat{v}_t$:

$$\begin{aligned} d||z_t||_{L_2} &= -\left(\frac{z_t A_t z_t}{||z_t||_{L_2}} + b_t \frac{z_t}{||z_t||_{L_2}}\right) dt \\ &\leq \left(-\lambda ||z_t||_{L_2} + b\right) dt. \end{aligned}$$

where $A_t := \frac{1}{n} \sum_i \int_0^1 \mathbb{I}_{(v_t-(1-s)(\hat{v}_t-v_t))x_i \geq 0} ds \ x_i x_i^T$ and $b_t := \frac{1}{n} \sum_i x_i \mathbb{I}_{x_i v_i \geq 0} B_{\epsilon,i} \leq \epsilon C_1 \frac{1}{n} \sum_i ||x_i|| =: b$. Note under the assumptions that the data is spherically distributed as in Theorem 6.4 in [23] there is some data depending constant $\lambda$ such that $zA_t z \geq \lambda ||z||_{L_2}^2$ for all $t > 0$ with high probability (sufficient data samples). This relies on the fact that $A_t$ is positive semi definite and that $A_t = 0$ iff $z_t = 0$. Then by Gronwall's lemma we have $||z_t||_{L_2} \leq \frac{b}{\lambda} \leq C\epsilon$. Therefore, for sufficiently small $\epsilon$ the trajectories stay close to each other. This implies the perturbed gradient flow enters the region of $v$ leading to convergence to the nearby stationary point $v^*$. $\square$

Note that the assumption $\max\{|a_i|^2, |a_r|^2\} \leq \epsilon$ in the above theorem is justified because $f$ learns a representation of $f_{\text{teacher}}(x)$ when solving a pre-training task where at least one of the neurons is effectively pushed to $0$ (as the teacher consists also of fewer neurons than the trained model $f$). Another way of dealing with the perturbation is to assume that there exist two neurons with the same $w$ and opposite signs. These two neurons would not contribute to the representation as they cancel each other out, thus no perturbation is incurred. Then we can remove the neuron that does not match the sign of $\tilde{a}$ and train with the other. This allows for an immediate application of Theorem 6.4 [23]. In our toy setup, we train with both as we do not assume to know the correct sign of the added neuron.

**Task difficulty measure for two-layer network**    The theoretical measure of how difficult our post training task is is captured by the distance in the function space $L_2(\mathbb{R}^d, p)$ between a reference task $\tilde{f}$ and final representation $f$. Concretely the measure is defined as

$$A_{\text{task}}(f, \tilde{f}) := ||f - \tilde{f}||_{L_2(\mathbb{R}^d, p)}^2 = \int_D \left| \sum_i a_i \sigma(w_i^T x) - \sum_i \tilde{a}_i \sigma(\tilde{w}_i^T x) \right|^2 dp(x)$$

where $p$ is a probability measure on the data space. The distance measure can be approximated by the use of the empirical measure and or an upper bound solely depending on the weight space. In the main text we assumed that we learned the representation $f_{\text{pre}}$ and that the finetuning task is given by $f_{\text{ft}} = f_{\text{pre}} + f_{\text{extra}}$

**Lemma E.2** *Denote the weights $a_i \in \mathbb{R}$ and $w_i \in \mathbb{R}^d$ for $i \in [n]$ of $f_{extra}$ and $p$ is a Gaussian with mean $\mu = 0$ and covariance matrix $\Sigma = I$. Then*

$$A_{task}(f_{ft}, f_{pre}) \leq \sum_i |a_i|^2 + ||w_i||^2$$

Proof. (1) Apply the triangle inequality neuron wise (since we have learned the teacher representation. (2) Gaussian integral calculations for a ReLU activation. (3) Apply Cauchy-Schwarz inequality to each $a_i|w_i|$.

Lemma E.2 substantiates the use of the $\ell_2$ distance in our toy example. Note the bound is tight in the balanced case.

**Single neuron fine turning**    We report in Table 8 here the test loss and the distance traveled in $\ell_2$ for the experiments in Figure 3.

Table 8: Average test loss on finetuning task and distance from pretrained initialization over 10 seeds. Small weights leads to better generalization and move less from the original representation.

| Algorithm | Test Loss | $\ell_2$ Distance |
|---|---|---|
| Small Weights | $0.0094 \pm 0.012$ | $0.027 \pm 0.0040$ |
| Large Gradient | $0.017 \pm 0.015$ | $0.057 \pm 0.032$ |

**More neurons fine turning**    We repeat the same experiment as in the main text but with an additional neuron. In Table 9 we observe that the variance for the distance by selecting the large gradients becomes high. This is in line what is observed in Figure 7c where we learn a complete new representation.

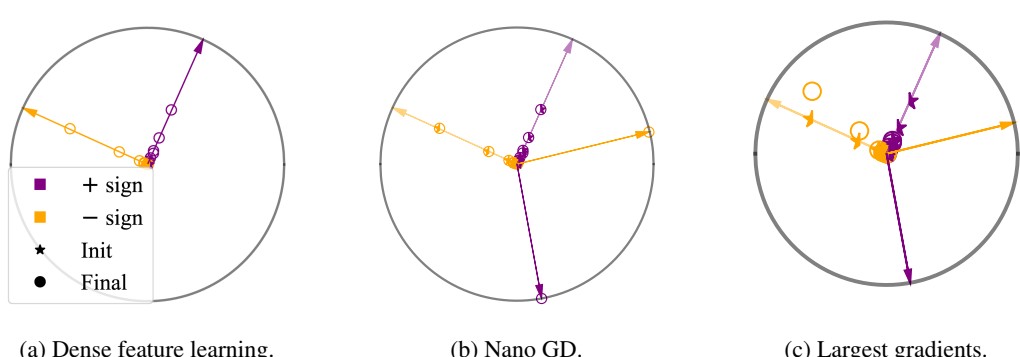

(a) Dense feature learning.    (b) Nano GD.    (c) Largest gradients.

Figure 7: nano gradient descent provably prevents catastrophic forgetting. (a) Two layer student network learns teach networks representation. (b) Nano gradient descent keeps the original representation while learning the two extra neuron. (c) The largest gradients can lead to learning completely different representations when the task transferability is high.

Table 9: Average test loss on finetuning task and distance from pretrained initialization over 10 seeds. Small weights leads to better generalization and move less from the original representation even when a more difficult representation needs to be learned i.e. less transferable tasks.

| Algorithm | Test Loss | $\ell_2$ Distance |
|---|---|---|
| Small Weights | $0.038 \pm 0.040$ | $0.042 \pm 0.0077$ |
| Large Gradient | $0.063 \pm 0.033$ | $0.097 \pm 0.071$ |

## F  Ablation study

### F.1  Small vs. large vs. random weights

For all configurations, we use a learning rate of $9 \times 10^{-5}$, weight decay of $0.0$, a batch size of 32, 5 training epochs, and a fixed random seed of 42. The mask density is initialized at $k_0 = 0.01$, the mask update interval is set to $m = 131$, and the density scheduler is disabled for this experiment. The training loss and evaluation metrics over steps are shown in Figure 8.

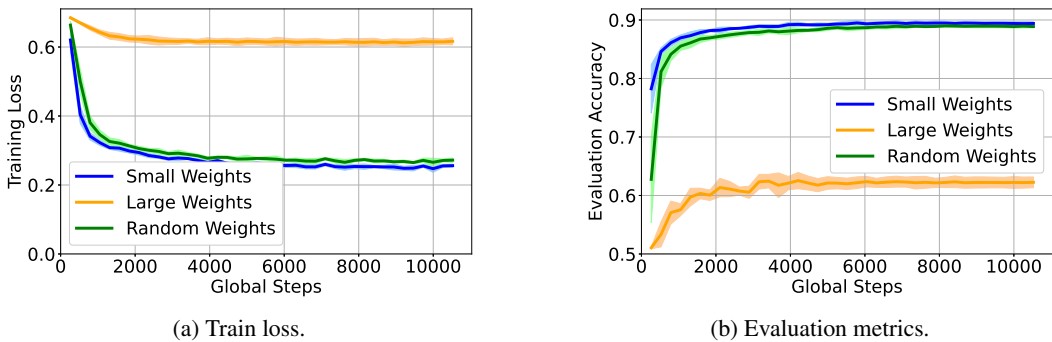

(a) Train loss.    (b) Evaluation metrics.

Figure 8: Ablation study comparing three masking strategies in NANOADAM: small-magnitude weights, large-magnitude weights, and random weights. Small-weight masking achieves the best training loss and evaluation performance under the same gradient density.

## F.2 Small weights vs. large gradients

**Study in NLP domain** We conduct an ablation study comparing two parameter selection strategies: (1) selecting parameters with the smallest absolute weight magnitudes, and (2) selecting parameters with the largest absolute gradient magnitudes. All experimental configurations follow the setup described in Appendix F.1, with the exception that the learning rate is separately tuned for each strategy to their best performance. The optimal learning rate is $1 \times 10^{-3}$ for small weights and $3 \times 10^{-4}$ for large gradients. The corresponding training loss and generalization performance are visualized in Figure 9.

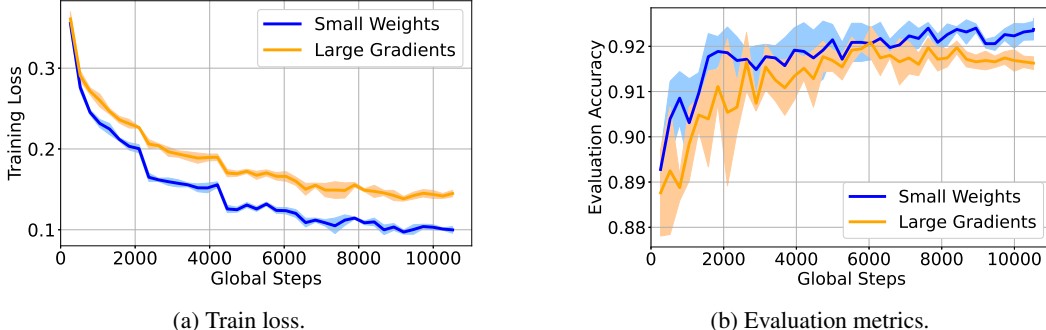

(a) Train loss.                (b) Evaluation metrics.

Figure 9: Ablation study comparing two selection criteria: small weights and large gradients. Small weight is better at generalization and convergence.

**Study in vision domain** We also provide an ablation study on FT CV tasks, where we compare NANOADAM under two different masking strategies: (1) Large gradients: selecting parameters with the largest absolute gradient magnitudes. (2) Small weights: selecting parameters with the smallest absolute weight magnitudes. Specifically, we finetune the ViT-Large model on the Flowers102 dataset, using the same hyperparameter settings detailed in Table 10. We initialize the mask density at $k_0 = 0.001$, and turn the density scheduler off. We set the mask update interval to $m = 100$. The resulting training loss and evaluation accuracy are presented in Figure 10. Note that in this experiment, we do not perform learning rate search for both strategies. Instead, we keep the same learning rate for both cases. Although the final performance is similar with both strategies, small weights achieve faster convergence.

Table 10: Hyperparameters for fully finetuning ViT-Large on Flowers102.

| LR | weight decay | batch size | epoch | seed |
|---|---|---|---|---|
| $3e-3$ | 0.0 | 128 | 10 | 42 |

## F.3 Dynamic mask vs. static mask

To evaluate the effectiveness of dynamic masking, we conduct experiments similar to the previous setup, with the key difference being the masking strategy used in NANOADAM. Specifically, for all configurations, we use a learning rate of $9 \times 10^{-5}$, weight decay of 0.0, a batch size of 32, 5 training epochs, and a fixed random seed of 42. The mask density is initialized at $k_0 = 0.01$, and the density scheduler is disabled for this experiment. We compare two approaches: (1) Dynamic masking, where the mask is updated every $m = 131$ steps; and (2) Static masking, where a fixed mask from the beginning is applied throughout the entire training process. The resulting training loss and evaluation accuracy are shown in Figure 11. While the static mask achieves very similar evaluation performance during the initial phase of training, the dynamic mask continues to improve and ultimately surpasses the static strategy in both evaluation accuracy and final training loss. These results indicate that dynamic masking allows the model to adapt more effectively throughout training, leading to better convergence and generalization.

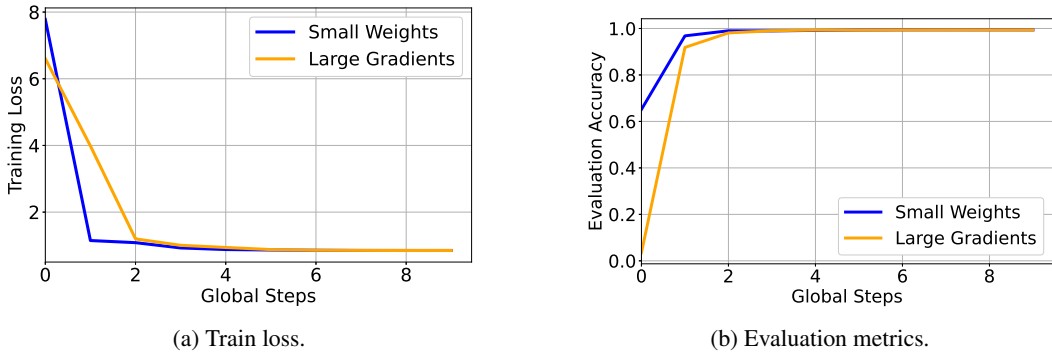

(a) Train loss.

(b) Evaluation metrics.

Figure 10: Ablation study on ViT-Large finetuned on the Flowers102 dataset comparing two selection criteria: small weights and large gradients. Small weight is better at generalization and convergence.

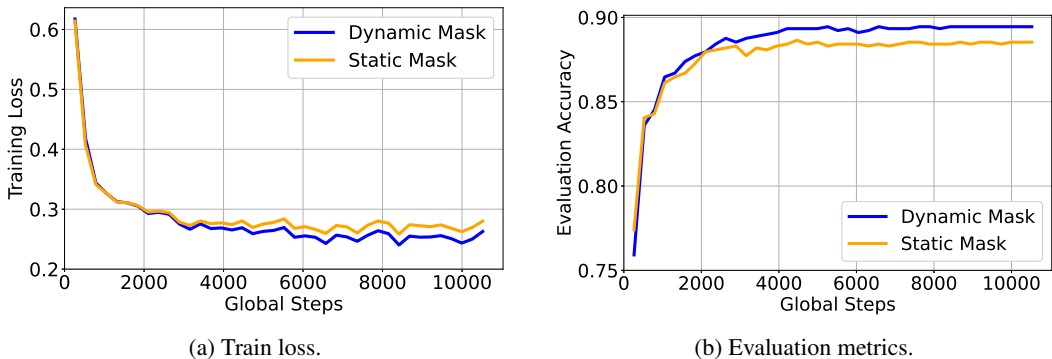

(a) Train loss.

(b) Evaluation metrics.

Figure 11: Ablation study on BERT-base finetuned on the SST2 task comparing two masking strategies in NANOADAM: dynamic mask and static mask. Dynamic mask achieves the best performance.

### F.4 Sensitivity Analysis

We conduct a sensitivity analysis by fine-tuning the BERT-base model on the QNLI task from the GLUE benchmark using a batch size of 32 and a learning rate of $1.1e - 4$.

To assess the impact of the mask interval parameter $m$, we disable the density scheduler and vary $m$ within the range [10, 1500]. The corresponding evaluation performance is presented in the table 11 below. Our results show that model performance remains largely stable across this range, with optimal results observed when m falls between 80 and 300. Given that the total number of steps in one epoch is $S = 3274$ in our experiment, this tuning range corresponds to approximately $0.024S$ to $0.09S$.

Table 11: Evaluation accuracy (%) vs. $m$.

| m | 10 | 50 | 80 | 100 | 300 | 500 | 700 | 900 | 1100 | 1500 |
|---|---|---|---|---|---|---|---|---|---|---|
| Eval (%) | 90.72 | 90.85 | 90.99 | 91.21 | 91.21 | 91.09 | 91.09 | 91.16 | 91.03 | 90.98 |

For the sensitivity analysis of the density interval $d$, we fix the mask interval at $m = 81$ and vary $d$ within the range [100, 1000]. The resulting evaluation performance is presented in the table below. We observe that when $d$ exceeds 300, the performance remains largely stable, with the best performance achieved at $d = 400$. Given that the total number of steps in one epoch is $S = 3274$, this corresponds to a density interval greater than approximately $(300/3274)S = 0.092S$, with the optimal interval around $(400/3274)S = 0.122S$.

Table 12: Evaluation accuracy (%) vs. $d$.

| d | 100 | 200 | 300 | 400 | 500 | 600 | 700 | 800 | 900 | 1000 |
|---|---|---|---|---|---|---|---|---|---|---|
| Eval (%) | 90.94 | 90.68 | 91.34 | 91.58 | 91.32 | 91.27 | 91.23 | 91.29 | 91.43 | 91.21 |

# G   Theoretical memory and computation overhead comparison

We now compare the theoretical memory footprint of NANOADAM with other optimizers, including AdamW, AdamW-8bit, and MicroAdam, focusing specifically on the optimizer state memory. Following the setup in [24], we assume the total number of model parameters is denoted by $S$. We use the LLaMA-2 7B model as a concrete example to illustrate memory consumption.

**NanoAdam.**   NANOADAM stores the optimizer states $m$ and $v$ (first- and second-order momentums) only for the selected subset of parameters in `bfloat16` format, with each requiring 2 Bytes. Given a gradient density $k$, the number of updated parameters is $kS$. Therefore, the total memory required for the momentums is: 2 (states) $\times$ 2 (Bytes) $\times kS = 4kS$ Bytes. Additionally, NANOADAM stores the mask $I$ indicating the indices of selected parameters. Simply using 64-bit integers (`long`), each index takes 8 Bytes, resulting in: $8 \times kS = 8kS$ Bytes. Alternatively, if we store indices in `int16` format (2 Bytes), as done in MicroAdam, the total memory reduces to: $4kS + 2kS = 6kS$ Bytes. For finetuning LLaMA-2 7B ($S = 6.275$ B) with $k = 0.01$, NANOADAM requires $6 \times 0.01S = 0.3765$ GB.

**AdamW.**   Stores both $m$ and $v$ for all parameters in `bfloat16`, requiring $2 \times 2 \times S = 4S$ Bytes. For LLaMA-2 7B, this equals $4S = 25.1$ GB.

**AdamW-8bit.**   Stores $m$ and $v$ in 8-bit precision, needing $2 \times 1 \times S = 2S$ Bytes, which corresponds to $2S = 12.55$ GB for LLaMA-2 7B.

**MicroAdam.**   Stores 4-bit quantized error ($0.5S$ Bytes) and a sliding window of size $m$ for both parameter indices in `int16` and values in `bfloat16`. Each step in the window stores $kS$ parameters, leading to $0.5S + 4mkS$ Bytes. For $k = 0.01$, $m = 10$, MicroAdam requires $0.5S + 4 \times 10 \times 0.01S = 5.65$ GB to fientune LLaMA-2 7B.

In terms of computation, the bottom-k operation does not introduce significant overhead. In our implementation, bottom-k selection is performed independently on each parameter matrix in each layer (excluding the final output layer), rather than applied globally. Specifically, it proceeds as follows: For each weight matrix, we first flatten it and divide it into subgroups of parameters (chunks). We then apply the bottom-k selection within each subgroup. As bottom-k operation has a time complexity of $O(k \log k)$, which remains computationally feasible in practice.

# H   Details and more results for finetuning on GLUE benchmark

## H.1   Hyperparameters for finetuning on GLUE benchmark

We largely follow the hyperparameter settings established by [24] for finetuning on various GLUE tasks. Specifically, we finetune for 5 epochs with a per-device batch size of 8, a fixed random seed of 42, and no weight decay. Unless otherwise stated, we perform grid search over the learning rate values $\{1e-6, 3e-6, 5e-6, 7e-6, 1e-5, 3e-5, 5e-5, 7e-5\}$ for all optimizers and models.

**GaLore.**   For GaLore, we set the low-rank approximation rank to $r = 256$ and vary the SVD update interval $T \in \{20, 200\}$. In contrast to the original GaLore implementation, which tunes both the scale and learning rate, we fix the scale to 1 and augment the learning rate search space with $\{1e-4, 3e-4, 5e-4, 7e-4\}$.

**MicroAdam.**   For MicroAdam, we use a sliding window of $m = 10$ gradients and a sparsity level of $k = 1\%$, resulting in an effective gradient sparsity of $mk = 10\%$. The quantization bucket size is set to 64.

**Adam and Adam-8bit.** All general hyperparameter settings mentioned above are directly applied to both Adam and Adam-8bit baselines.

**NanoAdam.** For NANOADAM, we set the initial update density to $k_0 = 10\%$ and linearly decay it to $4\%$ by the end of training. As training small weights typically requires a larger learning rate, we search over a wider range: $[5e-6, 3e-4]$. Additional hyperparameters specific to NANOADAM are summarized in Table 13.

Table 13: Hyperparameters for NANOADAM across tasks.

| task | COLA | SST2 | MRPC | STSB | QQP | MNLI | QNLI |
|---|---|---|---|---|---|---|---|
| mask interval | 6 | 52 | 7 | 13 | 711 | 306 | 81 |
| density interval | 33 | 263 | 14 | 27 | 1423 | 1533 | 409 |

## H.2 Additional results for finetuning on GLUE benchmark

The peak memory usage and running time are reported in Table. 15 and 16. We also compare with Nanoadam using the static mask strategy, which is shown in Table. 14 to illustrate the benefits of dynamic masking.

Table 14: Performance (eval metric) on GLUE dataset, comparing static mask strategy with our NanoAdam.

| Model | Method | COLA | SST2 | MRPC | STSB | QQP | MNLI | QNLI | AVG. |
|---|---|---|---|---|---|---|---|---|---|
| BERT -BASE | NANOADAM | 60.87 | 93.46 | 88.48 | 89.98 | 90.67 | 84.30 | 91.76 | **85.65** |
| | NanoAdam(static mask) | 56.24 | 91.51 | 84.07 | 89.68 | 89.75 | 82.59 | 90.92 | 83.54 |
| BERT -LARGE | NANOADAM | 66.85 | 94.61 | 90.20 | 90.86 | 91.03 | 86.40 | 92.44 | **87.48** |
| | NanoAdam(static mask) | 59.07 | 92.66 | 84.31 | 89.86 | 90.33 | 85.39 | 91.78 | 84.77 |
| OPT -1.3B | NANOADAM | 67.69 | 96.45 | 87.99 | 91.00 | 91.33 | 88.24 | 92.75 | **87.92** |
| | NanoAdam(static mask) | 60.38 | 95.30 | 86.03 | 90.55 | 91.07 | 87.61 | 91.91 | 86.12 |

Table 15: Memory usage (GB) on GLUE dataset.

| Model | Method | COLA | SST2 | MRPC | STSB | QQP | MNLI | QNLI | AVG. |
|---|---|---|---|---|---|---|---|---|---|
| BERT -BASE | Microadam | 3.64 | 3.63 | 3.64 | 3.64 | 3.81 | 3.75 | 3.79 | 3.70 |
| | NANOADAM | 3.58 | 3.60 | 3.59 | 3.57 | 3.60 | 3.60 | 3.59 | **3.59** |
| | Galore | 4.06 | 4.05 | 4.06 | 4.06 | 4.05 | 4.05 | 4.05 | 4.06 |
| | AdamW-8b | 3.72 | 3.72 | 3.72 | 3.72 | 3.72 | 3.72 | 3.72 | 3.72 |
| | AdamW | 3.94 | 3.94 | 3.95 | 3.95 | 3.94 | 3.93 | 3.95 | 3.94 |
| BERT -LARGE | Microadam | 5.56 | 5.53 | 5.52 | 5.54 | 5.54 | 5.53 | 5.54 | 5.54 |
| | NANOADAM | 5.21 | 5.24 | 5.15 | 5.19 | 5.23 | 5.22 | 5.19 | **5.20** |
| | Galore | 6.12 | 5.60 | 6.11 | 6.10 | 5.90 | 5.89 | 5.90 | 5.94 |
| | AdamW-8b | 5.61 | 5.83 | 5.62 | 5.61 | 5.61 | 5.62 | 5.60 | 5.64 |
| | AdamW | 6.45 | 6.52 | 6.47 | 6.47 | 6.52 | 6.47 | 6.46 | 6.48 |
| OPT -1.3B | Microadam | 13.20 | 13.15 | 13.19 | 13.19 | 13.19 | 13.19 | 13.20 | 13.19 |
| | NANOADAM | 11.33 | 11.76 | 11.41 | 12.04 | 11.66 | 11.77 | 11.57 | **11.65** |
| | Galore | 14.18 | 14.18 | 14.39 | 14.26 | 14.18 | 14.18 | 14.17 | 14.22 |
| | AdamW-8b | 13.07 | 13.08 | 13.08 | 13.08 | 13.08 | 13.08 | 13.08 | 13.08 |
| | AdamW | 18.18 | 18.16 | 18.16 | 18.16 | 18.17 | 18.17 | 18.16 | 18.16 |

Table 16: Training time (minutes) on GLUE dataset.

| Model | Method | COLA | SST2 | MRPC | STSB | QQP | MNLI | QNLI | AVG. |
|-------|--------|------|------|------|------|-----|------|------|------|
| BERT -BASE | Microadam | 3.25 | 17.79 | 1.78 | 6.26 | 98.79 | 103.42 | 28.41 | 37.10 |
| | NANOADAM | 2.99 | 16.57 | 1.46 | 2.09 | 91.35 | 94.05 | 26.72 | 33.60 |
| | NanoAdam(static mask) | 2.78 | 14.56 | 1.55 | 2.10 | 88.57 | 84.02 | 23.75 | 31.05 |
| | Galore | 2.01 | 12.16 | 0.86 | 1.51 | 72.65 | 69.99 | 19.08 | 25.47 |
| | AdamW-8b | 1.40 | 8.04 | 0.66 | 1.09 | 57.95 | 45.96 | 15.81 | 18.70 |
| | AdamW | 1.13 | 7.38 | 0.55 | 0.88 | 43.63 | 39.14 | 10.93 | 14.80 |
| BERT -LARGE | Microadam | 7.26 | 37.30 | 2.85 | 4.47 | 170.67 | 175.73 | 54.80 | 64.73 |
| | NANOADAM | 4.36 | 28.99 | 2.33 | 3.23 | 157.40 | 164.44 | 47.97 | 58.39 |
| | NanoAdam(static mask) | 4.71 | 25.01 | 2.61 | 3.49 | 146.48 | 150.09 | 39.59 | 53.14 |
| | Galore | 4.36 | 26.72 | 1.87 | 3.27 | 162.00 | 160.34 | 44.90 | 57.64 |
| | AdamW-8b | 3.88 | 18.85 | 1.08 | 1.79 | 96.64 | 93.20 | 25.38 | 34.40 |
| | AdamW | 2.04 | 12.91 | 0.93 | 1.56 | 78.37 | 78.07 | 21.16 | 27.86 |
| OPT -1.3B | Microadam | 9.17 | 46.14 | 5.55 | 7.10 | 238.46 | 253.18 | 70.35 | 89.99 |
| | NANOADAM | 4.95 | 37.12 | 2.54 | 4.23 | 186.92 | 196.63 | 53.74 | 69.45 |
| | NanoAdam(static mask) | 7.10 | 30.72 | 4.73 | 8.52 | 171.66 | 167.67 | 47.67 | 62.58 |
| | Galore | 12.50 | 84.24 | 4.85 | 8.37 | 451.76 | 463.27 | 129.09 | 164.87 |
| | AdamW-8b | 3.91 | 29.39 | 1.74 | 2.88 | 160.03 | 158.81 | 44.31 | 57.30 |
| | AdamW | 3.38 | 27.73 | 4.27 | 2.53 | 138.25 | 135.58 | 38.54 | 50.04 |

## H.3 Training Dynamics for finetuning NLP task

We also present the training dynamics observed in an NLP finetuning task. Specifically, we analyze the training loss and generalization performance of various optimizers when finetuning a BERT-base model on the QNLI task from the GLUE benchmark. For each optimizer, we perform hyperparameter tuning to determine the optimal learning rate. The experiment settings are the same as those described in Appendix H.1. The best learning rates for each method are summarized in Table. 17.

The resulting training loss and evaluation accuracy over time are shown in Figure 12a and 12b. As expected, in the early training steps, full finetuning with AdamW achieves the lowest training loss, since it updates the entire parameter set. However, at later stages, NANOADAM surpasses AdamW in both training loss and generalization. This is because updating only small-magnitude weights initially has minimal impact on the model output—dominated by large weights—but gradually exerts greater influence as training progresses.

In terms of generalization performance, AdamW performs better in the initial steps but is soon overtaken by NANOADAM, which consistently achieves higher accuracy in the later stages. Furthermore, across all training steps, NANOADAM outperforms MicroAdam, demonstrating its superior learning dynamics.

We also visualize the dynamics of the ratio between the number of parameters updated at least once and the total number of parameters during finetuning on the CoLA task using BERT-base. The results are shown in Figure 12c. Note that optimizers such as GaLore, AdamW, and AdamW-8bit update all parameters by design; thus, their curves are omitted for clarity. As shown, MicroAdam eventually updates over 90% of the parameters, whereas NANOADAM keeps more than 80% of parameters untouched throughout training.

Table 17: learning rate for various optimiser on finetuning Bert-base on QNLI.

| optimiser | AdamW | AdamW-8b | GaLore | MicroAdam | NANOADAM |
|-----------|-------|----------|--------|-----------|----------|
| LR | $7e-5$ | $7e-5$ | $1e-4$ | $4e-5$ | $1.1e-4$ |

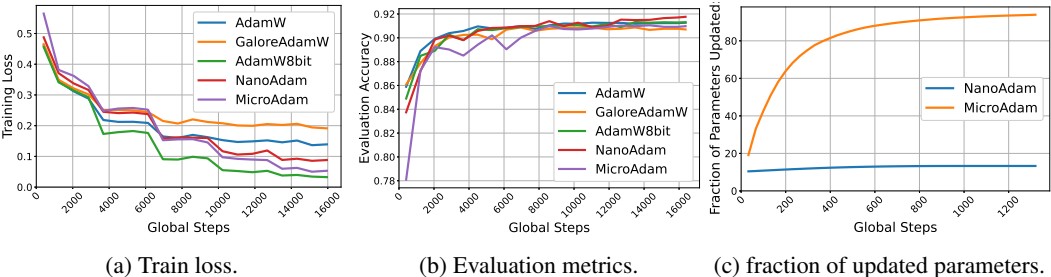

|                |                |                                    |
| (a) Train loss. | (b) Evaluation metrics. | (c) fraction of updated parameters. |

Figure 12: Dynamics of NLP FT.

# I    Details and more results for experiments on CV Tasks

## I.1    Details of Experiments on CV Tasks

**ViT-Large**    The detailed training configurations for the ViT-Large model are summarized in Tables 18–19, including both common and optimizer-specific hyperparameters. Task1 is CIFAR10 and task2 is Flowers102.

Table 18: Common hyperparameters used for finetuning ViT-Large.

| Batch Size | Seed | Weight Decay | LR Scheduler | Label Smoothing |
|---|---|---|---|---|
| 128 | 42 | 0.0 | CosineAnnealingLR | 0.1 |
| Epochs Task 1 | Epochs Task 2 | $\beta$ | $\epsilon$ | – |
| 5 | 5 | (0.9, 0.999) | $1 \times 10^{-8}$ | – |

Table 19: Optimizer-specific hyperparameters for ViT-Large.

| Optimizer | LR Task1 | LR Task2 | $k\,/\,k_0$ | Dynamic Density / $m$ | Mask Interval |
|---|---|---|---|---|---|
| NANOADAM | 1e-3 | 2e-3 | 0.1% | off | 100 |
| MicroAdam | 1e-4 | 1e-3 | 0.1% | $m = 10$ | – |
| AdamW | 1e-4 | 1e-4 | – | – | – |

**ResNet101**    The experimental settings for ResNet101 are summarized in Tables 20–21. These include common training hyperparameters and optimizer-specific configurations. Task1 is CIFAR10 and task2 is Flowers102.

Table 20: Common hyperparameters for ResNet101.

| Batch Size | Seed | Weight Decay | LR Scheduler | Label Smoothing |
|---|---|---|---|---|
| 128 | 42 | 0.0 | None | 0.0 |
| Epochs Task1 | Epochs Task2 | $\beta$ | $\epsilon$ | - |
| 30 | 30 | (0.9, 0.999) | 1e-8 | - |

**ResNet18**    For ResNet18, we use the same common settings as in Table 20, while the optimizer-specific hyperparameters for ResNet18 are summarised in Table. 22. Task1 is CIFAR10 and task2 is Flowers102.

Table 21: Optimizer-specific hyperparameters for ResNet101.

| Optimizer | LR Task1 | LR Task2 | $k / k_0$ | Dynamic Density / $m$ | Mask Interval |
|---|---|---|---|---|---|
| NANOADAM | 1e-2 | 7e-3 | 1% | off | 100 |
| MicroAdam | 1e-3 | 5e-3 | 0.1% | $m = 10$ | – |
| AdamW | 1e-3 | 1e-3 | – | – | – |

Table 22: Optimizer-specific hyperparameters for ResNet18.

| Optimizer | LR Task1 | LR Task2 | $k / k_0$ | Dynamic Density / $m$ | Mask Interval |
|---|---|---|---|---|---|
| NANOADAM | 9e-3 | 7e-3 | 1% | off | 100 |
| MicroAdam | 1e-3 | 5e-3 | 1% | $m = 10$ | – |
| AdamW | 1e-3 | 1e-3 | – | – | – |

## I.2 Dynamics of catastrophic forgetting

The generalisation performance of various optimisers over different tasks are shown in Figure 13- 15, while the experiment settings are reported in Appendix I.1.

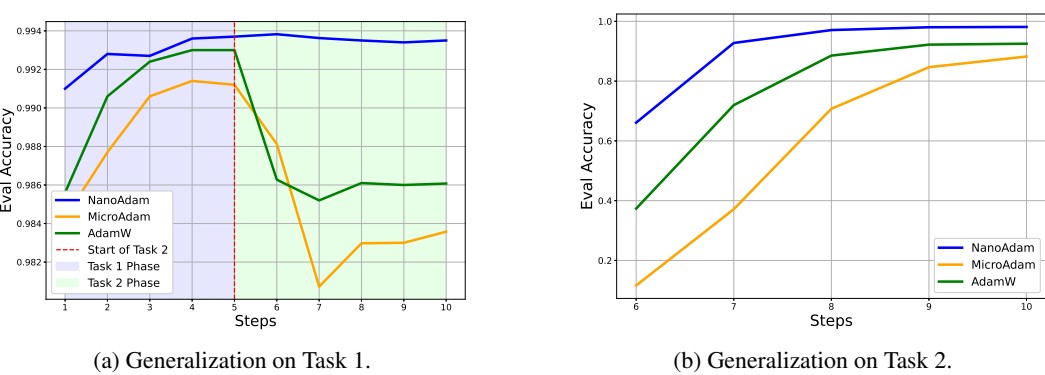

(a) Generalization on Task 1.          (b) Generalization on Task 2.

Figure 13: Catastrophic forgetting with ViT-Large.

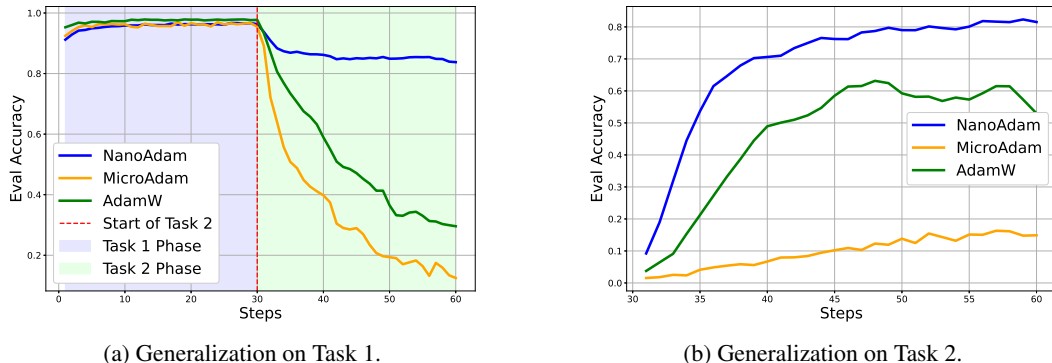

(a) Generalization on Task 1.          (b) Generalization on Task 2.

Figure 14: Catastrophic forgetting analysis on ResNet101. (a) Accuracy on Task 1 drops significantly after switching to FT on Task 2 for AdamW and MicroAdam, while NANOADAM retains performance. (b) NANOADAM also achieves better adaptation on Task 2.

## I.3 Parameter shift visualisation

To provide finer-grained insights, we visualize the layer-wise differences between the parameters of the pretrained ViT-Large model and those after continual learning on CIFAR10 and Flowers102, pre-

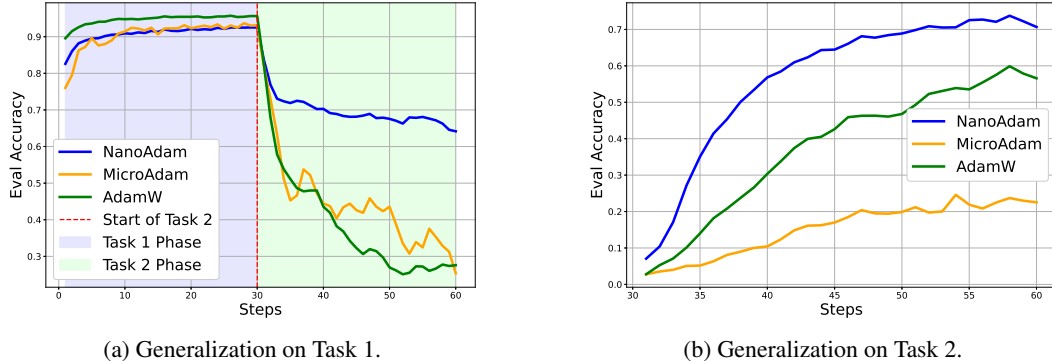

(a) Generalization on Task 1.

(b) Generalization on Task 2.

Figure 15: Catastrophic forgetting with ResNet18.

sented as heatmaps in Figure 16. The experimental setup follows the details reported in Appendix I.1. Notably, NANOADAM induces minimal drift in the attention weights (QKV), highlighting its stability in preserving critical features during continual learning.

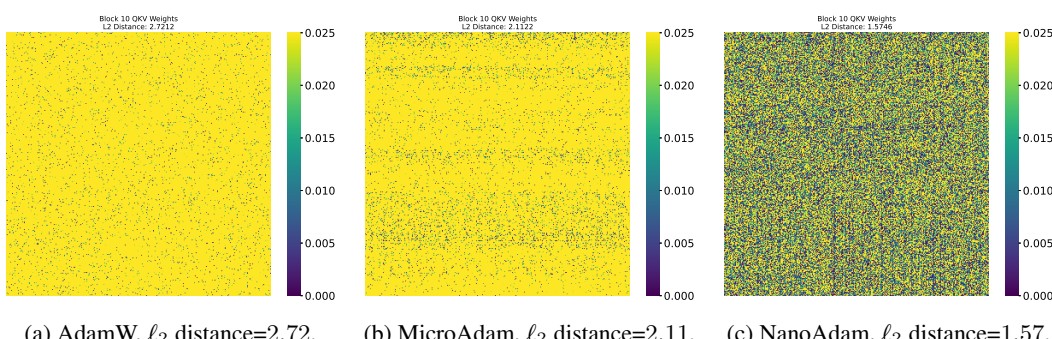

(a) AdamW, $\ell_2$ distance=2.72.     (b) MicroAdam, $\ell_2$ distance=2.11.     (c) NanoAdam, $\ell_2$ distance=1.57.

Figure 16: Parameter shift (qkv block) in Layer 10 after continual finetuning on CIFAR10 and Flowers102.

## I.4 Larger learning rate

We further investigate whether NANOADAM enables stable optimization under larger learning rates. We finetune ViT-Large (pretrained on ImageNet) on CIFAR-10 using NANOADAM, MicroAdam, and AdamW, representing finetuning of small weights, large gradients, and all weights respectively.

Using the same hyperparameter settings as in Tables 18–19, we vary the learning rate in $\{1e-5, 1e-4, 1e-3\}$. The resulting performance is shown in Figure 17. We observe that both AdamW and MicroAdam perform best at $1e-4$ and degrade significantly at $1e-3$. In contrast, NANOADAM can benefits from the larger learning rate, achieving its best performance at $1e-3$. This highlights its stability and effectiveness in aggressive optimization regimes.

## I.5 Effective learning rate

Effective Learning Rate (ELR) quantifies the actual rescaling applied to parameter updates during optimization. In adaptive optimizers such as Adam and its variants, the update rule incorporates element-wise adaptation based on the historical statistics of gradients. For a parameter $w$ at step $t$, the update is given by:

$$w_{t+1} = w_t - \eta \cdot \frac{\hat{m}_t}{\sqrt{\hat{v}_t} + \epsilon}$$

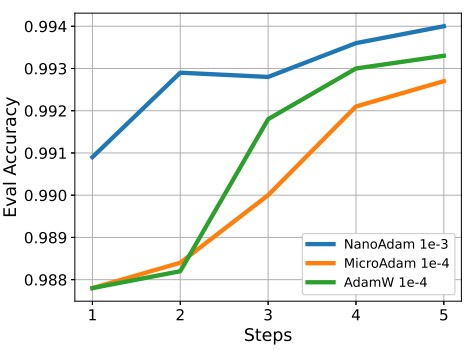
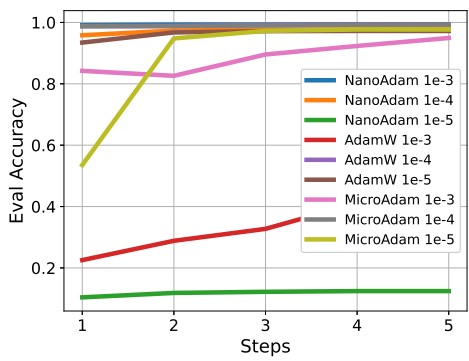

(a) performance with best learning rate.

(b) performance with full learning rate grid.

Figure 17: Small weights allow large learning rate.

Here, $\eta$ denotes the global learning rate; $\hat{m}_t$ and $\hat{v}_t$ represent the bias-corrected first and second moment estimates, respectively; and $\epsilon$ is a small constant added for numerical stability. The *effective learning rate* is thus defined as:

$$\text{ELR} = \frac{\eta}{\sqrt{\hat{v}_t} + \epsilon}$$

Since ELR is computed per-parameter and evolves over time, its magnitude provides insight into how aggressively each parameter is being updated. For sparse optimizers such as NANOADAM and MicroAdam, which selectively update a subset of parameters, we compute ELR only over the actively updated parameters. The final reported metric is the average ELR across these selected parameters. We present the ELR dynamics of NANOADAM, MicroAdam, and AdamW during finetuning on both vision and language tasks.

For the computer vision task, we finetune the ViT-Large model (pretrained on ImageNet) on CIFAR10 using various optimizers. The experimental settings follow those described in Tables 18–19. The ELR trends for different optimizers are depicted in Figure 18. For the NLP task, we finetune the BERT-base model on the SST-2 dataset from the GLUE benchmark. The experimental details are provided in Appendix H.1. As shown in the results, NANOADAM enables a more aggressive effective learning rate compared to other methods.

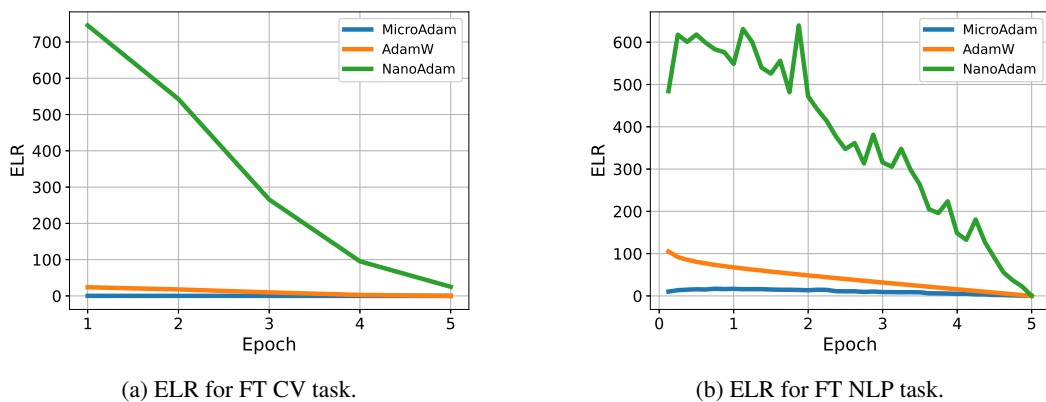

(a) ELR for FT CV task.

(b) ELR for FT NLP task.

Figure 18: Comparison of effective learning rate in FT CV and NLP task.

## J    Details and results for finetuning on Commonsense benchmark

We evaluate our method, NANOADAM, alongside MICROADAM and AdamW on fully fine-tuning the Llama 3.2 3B model on the Commonsense Benchmark. For these experiments, we exclude the final

`lm_head` and all normalization layers from partial updates; these layers are fully updated. We largely follow the experimental setup provided by the DoRA repository and utilize gradient checkpointing. The benchmark consists of eight tasks: HellaSwag, Winogrande, PIQA, ARC-Easy, ARC-Challenge, OpenBookQA, SocialIQA, and BoolQ. All models are trained on a compute node equipped with $4\times$A100 40GB GPUs. The training procedures are kept identical across methods, with the exception of individually tuned learning rates. We note that MICROADAM diverges when using a learning rate larger than $1e-5$. The hyperparameters used for these experiments are summarized in Tables 23 and 24. Note that both MICROADAM and NANOADAM use an effective gradient sparsity of 90%.

It's important to note that our proposed method functions as an efficient optimizer, and is thus orthogonal to LoRA—it can be combined with LoRA rather than serving as an alternative. Therefore, we also conduct the same experiments with LoRA (r=32), which is trained using AdamW and NanoAdam, respectively. For fientuning LoRA with AdamW, we use learning rate of $1e-4$, and for fientuning LoRA with NanoAdam, we use a learning rate of $7e-4$, density $k=10\%$ and mask interval of 600 and turn off the dynamic density.

In Table 25, we report the performance on each task, the average performance, as well as the training time and average memory usage across 4 GPUs. As shown, NANOADAM achieves the best average accuracy while also reducing memory consumption compared to the other methods.

Table 23: Common hyperparameters used for finetuning Llama3.2 3B on Commonsense benchmark.

| Seed | Batch Size | micro batch size | epochs | cutoff length | $\beta$ | $\epsilon$ |
|------|-----------|-----------------|--------|--------------|---------|-----------|
| 42 | 16 | 2 | 2 | 256 | (0.9, 0.999) | $1 \times 10^{-8}$ |

Table 24: Optimizer-specific hyperparameters for finetuning LLaMA3.2-3B on Commonsense tasks.

| Optimizer | LR | $k$ / $k_0$ | Dynamic Density / $m$ | Mask Interval |
|-----------|-----|-----------|----------------------|--------------|
| NANOADAM | 6e-5 | 10% | off | 600 |
| MicroAdam | 1e-5 | 1% | $m$ = 10 | – |
| AdamW | 1e-5 | – | – | – |

Table 25: Performance of fine-tuning the Llama 3.2 3B model on the Commonsense Benchmark: average accuracy (%), memory usage (GB) and time (h).

| Method | HellaSwag | Winogrande | PIQA | ARC-Easy | ARC-Challenge | OpenBookQA | SocialIQA | BoolQ | AVG. | memory | time (h) |
|--------|-----------|-----------|------|---------|--------------|-----------|-----------|-------|------|--------|---------|
| MicroAdam | 92.44 | 83.27 | 85.91 | 85.82 | 73.38 | 81.40 | 79.07 | 63.94 | 80.65 | 28.64 | 12.25 |
| NANOADAM | 93.21 | 82.48 | 86.07 | 85.86 | 74.91 | 82.20 | 79.99 | 71.71 | **82.05** | **25.30** | **10.11** |
| AdamW | 77.71 | 74.11 | 63.33 | 83.63 | 68.34 | 75.20 | 76.10 | 66.67 | 73.14 | 37.75 | 21.8 |
| LoRA (r=32) NanoAdam | 88.19 | 80.35 | 83.03 | 84.30 | 70.39 | 79.60 | 77.48 | 69.42 | **79.09** | **11.45** | **3.13** |
| LoRA (r=32) AdamW | 88.38 | 80.43 | 84.22 | 84.89 | 72.35 | 80.20 | 79.02 | 61.53 | 78.88 | 15.24 | 2.84 |

## K  Details and results for finetuning on GSM-8k benchmark

We now validate the effectiveness of various optimization methods on a finetuning task. Specifically, we finetune `LLaMA2-7B` on the GSM-8k dataset, a challenging benchmark for grade-school-level mathematical reasoning. Our experiments largely follow the codebase and the settings of MicroAdam [24].

The model is trained for 3 epochs with a global batch size of 32. The micro-batch size per device is set to `auto`, and the maximum input sequence length is 512. To ensure robustness, we run experiments across four random seeds: $\{7, 42, 100, 512\}$. The hyperparameter configurations for each method are summarized in Table 26, and the corresponding results are reported in Tables 27 and 28. It is worth noting that while the configured density levels $k$ or $k_0$ vary across methods, the resulting effective gradient density remains approximately 10% for all.

As shown in the results, NANOADAM outperforms both AdamW8b and MicroAdam in terms of accuracy, while also reducing memory usage and maintaining a runtime comparable to AdamW8b.

Table 26: Optimizer-specific hyperparameters for finetuning LLaMA2-7B on GSM-8k tasks.

| Optimizer | LR | $k$ / $k_0$ | Dynamic Density / $m$ | Mask Interval | betas |
|---|---|---|---|---|---|
| NANOADAM | 4e-4 | 10% | off | 5 | $(0.75, 0.999)$ |
| MicroAdam | 4e-5 | 1% | $m = 10$ | – | $(0.9, 0.999)$ |
| AdamW8b | 4e-5 | – | – | – | $(0.9, 0.999)$ |

Table 27: Accuracy comparison of finetuning LLama2-7B using various optimisers on GSM-8k tasks.

| Method | seed=7 | seed=42 | seed=100 | seed=512 | Mean | Std |
|---|---|---|---|---|---|---|
| AdamW8b | 33.28 | 34.27 | 33.36 | 34.04 | 33.74 | 0.49 |
| Microadam | 33.43 | 34.42 | 33.59 | 34.80 | 34.06 | 0.66 |
| NANOADAM | 34.27 | 34.57 | 35.63 | 35.33 | 34.95 | 0.64 |

Table 28: Total memory overhead and full run time of finetuning LLama2 7B using various optimisers on GSM-8k tasks. Results are averaged over 4 seeds.

| Method | run time (h) | memory (GB) |
|---|---|---|
| AdamW8b | 0.40 | 43.27 |
| Microadam | 0.47 | 38.90 |
| NANOADAM | 0.41 | 36.68 |

## L    Transfer across dissimilar domains

To evaluate the potential limitations of our method when transferring across dissimilar domains, we fine-tuned a ResNet-18 model pretrained on ImageNet (a general-domain dataset) on the PathMNIST task from the MedMNIST dataset—a medical image classification task with 9 classes. We compared the performance of full fine-tuning using AdamW (learning rate = 7e-4) with our method using NanoAdam (sparsity = $1\%$, learning rate = $1e-3$). AdamW achieve evaluation accuracy $90.85\%$ while NanoAdam achieves $90.63\%$. Despite the domain shift, NanoAdam achieves performance comparable to full fine-tuning, demonstrating its robustness even in cross-domain adaptation scenarios.

## M    Sensitivity to initial weight distribution

To evaluate sensitivity to the initial weight distribution, we conducted the following experiment: we started with a ResNet-18 model pretrained on ImageNet and pruned $80\%$ of the weights based on magnitude. We then fine-tuned the resulting sparse network on CIFAR-10 using both Adam (learning rate $1e-3$) and NanoAdam (gradient density=$1\%$, learning rate $5e-3$). Adam achieved an evaluation accuracy of $85.34\%$, while NanoAdam reached $86.16\%$, suggesting that NanoAdam is even more effective under sparse initialization. We hypothesize that this is because pruning removes $80\%$ of the small weights, leaving the network to rely primarily on large weights during full fine-tuning—a strategy we have shown to be inefficient in our ablation study. In this case, focusing updates on the remaining small weights, as NanoAdam does, leads to better adaptation and generalization.

