# OpenReview forum: "Pay Attention to Small Weights"
_NeurIPS.cc/2025/Conference — NeurIPS 2025 poster_

### Official Review · Reviewer_yCVR · 2025-06-09

**Clarity:** 2
**Significance:** 2
**Originality:** 2
**Rating:** 4
**Confidence:** 3

**Summary:**

The paper proposes NanoAdam, a gradient-free, memory-efficient optimizer for finetuning large pretrained models. Motivated by the empirical observation that small-magnitude weights often correspond to large gradients, the method restricts updates to low-magnitude weights. This approach is designed to reduce catastrophic forgetting, preserve pretraining features, and improve training efficiency.

**Questions:**

n/a

**Ethical Concerns:**

["NO or VERY MINOR ethics concerns only"]

**Final Justification:**

The authors' responses resolve most of my concerns, I would adjust my rating.

**Limitations:**

While the authors briefly address limitations in Section 2.2, the discussion primarily reads as an analysis of the method’s advantages rather than a critical evaluation of its shortcomings. It would strengthen the paper if the authors included a dedicated section explicitly discussing the limitations and broader implications of their approach, including aspects such as the generalizability of the optimizer, its computational efficiency, and potential trade-offs in different model architectures or training regimes.

**Quality:**

2

**Strengths And Weaknesses:**

- While the observed correlation between large gradients and small weights is interesting, the assumption that small weights are always the best candidates for updates is overly simplistic. In Adam-style optimizers, gradients are normalized by past variance; thus, even small weights with high gradients may not result in large updates. Moreover, weight magnitude is not always indicative of parameter importance, especially in transformers where scale and function can differ layer-wise.
- NanoAdam dynamically estimates the subset of smallest weights at intervals via sorting (Bottom_k(|θ|)). Although this occurs infrequently (every m steps), it could still introduce overhead, particularly in large models with tens of billions of parameters. It is unclear whether this cost counteracts the claimed efficiency gains—especially in practice on large-scale LLMs.
- Despite strong results on BERT and vision models, it’s unclear how NanoAdam scales to contemporary LLMs (e.g., Llama, Mistral) with model parallelism and pipeline layers. The dynamic masking may interfere with hardware-accelerated fused optimizers and distributed training frameworks, which are critical for LLM-scale deployment.
- The paper’s writing quality is suboptimal and affects readability. For instance, in the introduction (lines 16–38), the authors discuss motivation and related challenges, but abruptly switch to “Our method…” without first introducing what the method is or how it addresses the problem. Such transitions feel disjointed, and similar structural issues appear throughout the paper, weakening the overall presentation and clarity.

---

> ### Author Rebuttal · Authors · 2025-07-30
>
> We thank the reviewer for appreciating our insights as interesting and our empirical results as strong and giving us valuable feedback to improve our manuscript. Below, we address their insightful comments and answer their questions. We look forward to a potential exchange during the discussion period.
>
> ### 1. Small weights and large updates
>
> We appreciate the reviewer’s insight regarding the behavior of Adam-like optimizers and their gradient normalization effect. To better understand the actual parameter changes during fine-tuning, we conducted an additional experiment: we fine-tuned a BERT-base model on the CoLA task from the GLUE benchmark using AdamW for 5 epochs.
>
> For each layer, we partitioned the parameters into three groups based on their absolute magnitudes in the pretrained model:
>
> 1. The bottom 30% (smallest magnitudes)
> 2. The top 30% (largest magnitudes)
> 3. The remaining middle 40%
>
> We then measured the average change in weights within each group, calculated as the difference between the pretrained and fine-tuned weights. This analysis allows us to examine which groups of parameters receive the largest updates during fine-tuning.
>
> The results, presented in the table below, consistently show that across all layers, the smallest-magnitude weights undergo the largest updates, while the largest-magnitude weights change the least. This trend holds regardless of the layer’s scale or functional role in the Transformer architecture.
>
> Additionally, in our implementation, the bottom-k selection is performed per layer, not globally—ensuring layer-wise adaptivity.
>
> | **Layer** | **30% Smallest** | **30% Largest** | **Remaining** |
> | --------------- | ---------------------- | --------------------- | ------------------- |
> | Layer 0         | 0.000799               | 0.000381              | 0.000393            |
> | Layer 1         | 0.000815               | 0.000325              | 0.000363            |
> | Layer 2         | 0.000787               | 0.000331              | 0.000344            |
> | Layer 3         | 0.000812               | 0.000302              | 0.000353            |
> | Layer 4         | 0.000846               | 0.000298              | 0.000354            |
> | Layer 5         | 0.000877               | 0.000282              | 0.000351            |
> | Layer 6         | 0.000821               | 0.000294              | 0.000342            |
> | Layer 7         | 0.000798               | 0.000299              | 0.000341            |
> | Layer 8         | 0.000727               | 0.000276              | 0.000321            |
> | Layer 9         | 0.000691               | 0.000278              | 0.000308            |
> | Layer 10        | 0.000669               | 0.000280              | 0.000302            |
> | Layer 11        | 0.000742               | 0.000279              | 0.000324            |
>
> We agree that parameter magnitude is not always a perfect indicator of importance. In fact, for this reason, most of our analysis has served the purpose to understand under which conditions small weight magnitude is associated with high gradients (and thus high potential for adaptability to a new task) and when we can therefore expect our method NanoAdam to work well: We both require a high alignment between the pre-training and finetuning task (i.e. good transferability) and high degree of overparameterization. We will also highlight this fact in a separate limitations section in our updated manuscript. It is also noteworthy that our objective is not to identify the most precise weight importance measure. Rather, our aim is to develop a simple yet highly efficient criterion for finetuning that significantly reduces memory overhead from gradient updates while preserving—or even improving—generalization performance.
>
> ### 2. Scalability and overhead
>
>
> We appreciate the reviewer’s observation regarding the potential overhead introduced by the bottom-k operation. In our implementation, bottom-k selection is performed independently on each parameter matrix in each layer (excluding the final output layer), rather than applied globally. Specifically, it proceeds as follows: For each weight matrix, we first flatten it and divide it into subgroups of parameters (chunks). We then apply the bottom-k selection within each subgroup. As bottom-k operation has a time complexity of O(k \log k), which remains computationally feasible in practice.
>
> To assess scalability, we conducted experiments on larger models such as LLaMA 2-7B. The results of fine-tuning the LLaMA 2-7B model are shown in the table below. Despite the increase in model size, our method continues to deliver better generalization performance and significant memory savings, with only a modest increase in runtime overhead.
>
>
> | **Method** | **Eval Acc (%)** | **Memory (GB)** | **Runtime (h)** |
> | ---------------- | ---------------------- | --------------------- | --------------------- |
> | AdamW-8b         | 33.74                  | 43.27                 | 0.40                  |
> | MicroAdam        | 34.06                  | 38.90                 | 0.47                  |
> | NanoAdam         | 34.95                  | 36.68                 | 0.41                  |
>
> ### 3. Compatibility with modern techniques
>
> We appreciate the reviewer’s insight regarding the scalability of our method to contemporary large language models (e.g., LLaMA, Mistral), particularly in the context of model parallelism and pipeline layers. In our implementation, the bottom-k selection is applied independently to each parameter matrix within each layer, rather than performed globally across the model. This design ensures compatibility with modern hardware acceleration techniques and parallel training frameworks, and avoids conflicts with model or pipeline parallelism.
>
> ### 4. Writing
>
> We will thoroughly review the paper and revise the relevant sections to ensure smoother and less abrupt content switches. For instance, in the Introduction (lines 16–38), after outlining the motivation and related challenges, we will revise the transition sentence as follows: “In this work, we propose a method that relies purely on parameter magnitude—without the use of gradient information or error feedback—resulting in improved computational efficiency.”
>
>
> ### Limitations
>
> We will include a dedicated section discussing the limitations and broader implications of our method. This will address scalability and overhead concerns( as noted in points 2 and 3 above), clarify that the method is designed for the fine-tuning scenario and would likely not work as well in pretraining, as its success relies on knowledge transferability between pretraining and fine-tuning tasks. We will also highlight that the method benefits from model overparameterization, which is a key factor in its effectiveness. We will include a dedicated section discussing the limitations and broader implications of our method, as follows.
>
> Our proposed method NanoAdam introduces minimal computational overhead. Specifically, for each weight matrix in each layer, we first flatten the matrix and divide it into subgroups (chunks), then apply bottom-k selection within each subgroup. This process is applied uniformly across both convolutional and MLP layers. The main computational cost arises from the bottom-k operation, which has a time complexity of O(k \log k).
>
> Thanks to its layer-wise and parameter-wise design, the method is naturally scalable to larger models and remains compatible with modern hardware acceleration and parallel training frameworks. It avoids conflicts with model parallelism and pipeline layers, making it practical for contemporary large-scale architectures.
>
> However, NanoAdam the method has several limitations. Its effectiveness relies heavily on knowledge transferability and overparameterization. When the pretraining and finetuning tasks are well-aligned, the method helps avoid catastrophic forgetting and effectively leverages the plasticity of small weights to adapt to new tasks. In contrast, when there is limited similarity between tasks, the method may underperform compared to full-update optimizers like Adam.
> Moreover, the method benefits significantly from model overparameterization. As demonstrated in our experiments on vision tasks, scaling from a smaller model (e.g., ResNet-18) to a larger one (e.g., ViT-Large) results in improved overall performance and reduced forgetting. This suggests that NanoAdam the method is particularly well-suited for large, overparameterized models.

---

> > ### Comment · Reviewer_yCVR · 2025-08-05
> >
> > The authors' responses resolve most of my concerns, I would adjust my rating.

---

### Official Review · Reviewer_Vq6D · 2025-07-01

**Clarity:** 3
**Significance:** 4
**Originality:** 3
**Rating:** 4
**Confidence:** 3

**Summary:**

This paper introduces NANOADAM, a new optimization technique aimed at improving the efficiency and effectiveness of fine-tuning large pre-trained neural networks. Rather than selecting parameters for update based on gradient magnitude (a common practice), NANOADAM selects parameters with small absolute weight magnitudes, arguing that these are more “plastic” and less likely to encode critical pretrained knowledge.

The authors observe a consistent hyperbolic correlation between gradients and weight magnitudes during finetuning (particularly in NLP and vision tasks), i.e., large gradients tend to correspond to small weights. They propose leveraging this correlation to select and update only small weights dynamically. NANOADAM avoids the need for gradient-based selection or error feedback mechanisms, improving memory efficiency and preserving pretrained representations. Apparently, this reduces catastrophic forgetting.

Extensive experiments across NLP (GLUE) and CV (CIFAR-10 and Flowers102) tasks demonstrate that NANOADAM outperforms existing memory-efficient optimizers (MicroAdam, GaLore, AdamW-8bit) in terms of generalization performance, memory usage, and parameter shift.

**Questions:**

1. Often for FT we freeze a set of layers. How does the number of layers to freeze affect the framework? I am thinking simple convnets with a MLP on top. How do convolutional and MLP layers differ?

2. Could the authors provide further clarification or experimental results on why NANOADAM does not perform well when training from scratch? Is it primarily due to the lack of structure in weight initialization?

3. While LoRA is cited as a PEFT baseline, it was not included in the experiments. Could the authors justify this omission or include results in the appendix?

4. While the paper justifies excluding error feedback mechanisms, did the authors test simpler forms of feedback (e.g., residual accumulation without reinsertion) to see if performance improves marginally without major cost?

5.  How robust is NANOADAM to variations in m (mask interval) and d (density interval)? Some empirical sensitivity curves or robustness plots would be helpful.

6. NANODAM was proven for small parameter model. Do we trust this holds for large 7B+ models?

**Ethical Concerns:**

["NO or VERY MINOR ethics concerns only"]

**Limitations:**

They acknowledge that small weights do not always fully overlap with large gradients (Section 2.2), and that their method is explicitly designed for finetuning rather than training from scratch.

However, they could further expand on:

Potential pitfalls in transfer across highly dissimilar tasks (e.g., general-domain pretraining to low-resource medical tasks).

Sensitivity to initial weight distributions—e.g., would pretrained weights that are uniformly sparse or quantized change NANOADAM's effectiveness?

**Paper Formatting Concerns:**

None.

**Quality:**

4

**Strengths And Weaknesses:**

Strengths
Novel Insight into Sparse Finetuning: The paper presents a compelling observation—large gradients correlate with small weights—and builds a theoretically and empirically grounded argument for exploiting this in fine-tuning.

Through a teacher-student model, the paper provides a clean and illustrative theoretical explanation for why updating small weights can avoid catastrophic forgetting (Theorem 2.2), which supports the method’s design.

Efficiency Without Gradient Use: NANOADAM does not rely on gradients for mask selection, enabling precomputation and reducing memory/computation overhead compared to methods like MicroAdam or DST.

Clear Ablations: The ablation studies (Figure 4) directly compare small weights vs large weights vs random weights, and small weights vs large gradients, showing consistent gains for the proposed strategy.

Excellent Empirical Results:

Table 2 shows that NANOADAM performs best across nearly all tasks for BERT-base, BERT-large, and OPT-1.3B.

Table 3 shows lower memory usage.

Table 4 and 5 demonstrate its ability to reduce catastrophic forgetting and parameter drift during continual learning.

Broad Applicability: Evaluated across both NLP and CV tasks, with results on transformer and CNN architectures, making the claims more generalizable.

Weaknesses:
1. No Comparison to LoRA-type PEFT Methods: While LoRA is mentioned in the related work, no head-to-head comparisons are presented. Although LoRA adds parameters, it is a strong PEFT baseline and should have been benchmarked.

2. Density Scheduler Sensitivity: While the density scheduler and update intervals are mentioned, the method might be sensitive to these hyperparameters. The default choices and tuning ranges could have been discussed in more detail (e.g., effect of mask interval m on performance).

---

> ### Author Rebuttal · Authors · 2025-07-30
>
> We thank the reviewers for their valuable feedback and insightful comments. We are happy to address the concerns below.
>
> ### 1. Layer freezing
>
> To understand how freezing different layers affects our framework, we conducted an ablation study using ResNet-18 on CIFAR-10. We systematically froze each layer from the first to the final fully connected (FC) layer and evaluated generalization performance:
>
> | **Layer Frozen** | **Eval Acc (%)** |
> | ---------------------- | ---------------------- |
> | No freeze              | 92.58                  |
> | Freeze layer1          | 91.53                  |
> | Freeze layer2          | 91.29                  |
> | Freeze layer3          | 91.36                  |
> | Freeze layer4          | 91.22                  |
> | Freeze FC              | 91.51                  |
>
> We observe that freezing intermediate layers results in more severe degradation, while freezing early or final layers has less impact. This suggests that selectively updating parameters—rather than completely freezing entire layers—is more effective for preserving generalization in fine-tuning.
>
> ### 2. Training from scratch
>
> Our hypothesis is that the effectiveness of NanoAdam stems from: (1) Knowledge transferability and (2) Overparameterization.
>
> As shown in Table 4 of the manuscript, moving from smaller models (e.g., ResNet-18) to larger ones (e.g., ViT-Large)—i.e., increasing overparameterization— improves performance and reduces forgetting. This supports the view that NanoAdam benefits from the overparameterized nature of modern deep networks.
>
> To further examine the impact of Knowledge transferability, we trained ResNet-18 and ResNet-101 from scratch on CIFAR-10 and compared AdamW (LR=1e-3) with NanoAdam (90% sparsity, LR=5e-3). This setup can be considered an extreme case of transferring knowledge from a null domain (i.e., no pretraining) to a vision domain. The generalization results are presented in the table below. We observe that NanoAdam incurs about a 1% drop in accuracy compared to AdamW, suggesting that it is less effective in training from scratch or when there is a mismatch between the knowledge in pretraining and finetuning tasks. Notably, the more overparameterized ResNet-101 shows a smaller performance drop, further indicating that overparameterization contributes to the effectiveness of NanoAdam.
>
> | **Model** | **AdamW (%)** | **NanoAdam (%)** |
> | --------------- | ------------------- | ---------------------- |
> | ResNet-18       | 92.48               | 91.29                  |
> | ResNet-101      | 94.08               | 93.06                  |
>
> ### 3. Comparison with LoRA
>
> We thank the reviewer for raising the concern regarding comparisons with LoRA. In response, we have conducted experiments to directly compare our method with LoRA. It’s important to note that our proposed method functions as an efficient optimizer, and is thus orthogonal to LoRA—it can be combined with LoRA rather than serving as an alternative. Specifically, we fine-tuned a LLaMA 3.2-3B model on the Commonsense benchmark using the following setups:
>
> * Full fine-tuning with AdamW
> * Full fine-tuning with NanoAdam (90% sparsity)
> * LoRA (r=32) with AdamW
> * LoRA (r=32) with NanoAdam (90% sparsity)
>
> The results are summarized in the table below. We observe that full fine-tuning with NanoAdam yields the highest performance, albeit with increased memory usage compared to LoRA. However, when memory efficiency is a priority, combining LoRA with NanoAdam proves highly effective—reducing memory consumption from 15.2 GB to 10.7 GB while also achieving improved generalization performance.
>
> | **Method**          | **Eval Acc (%)** | **Memory (GB)** | **Runtime (h)** |
> | ------------------------- | ---------------------- | --------------------- | --------------------- |
> | Full FT with AdamW        | 73.14                  | 37.75                 | 21.80                 |
> | Full FT with NanoAdam     | 82.05                  | 25.30                 | 10.11                 |
> | LoRA (r=32) with AdamW    | 78.88                  | 15.24                 | 2.84                  |
> | LoRA (r=32) with NanoAdam | 79.09                  | 10.73                 | 3.13                  |
>
> ### 4. Error feedback mechanism
>
> We appreciate the reviewer’s insightful comment regarding the error feedback mechanism. We experimented with incorporating a similar error feedback strategy as used in MicroAdam; however, we observed that it actually degraded performance in our setting.
>
> We hypothesize that this is due to the nature of our parameter selection strategy, which is based on the magnitude of the weights themselves. As discussed in Section 2.2, our small-weights update rule does not aim to replicate the trajectory of large gradient updates. Instead, it follows a distinct and highly efficient learning tracetory in finetuning. Introducing gradient accumulation and feeding the residual error back into this trajectory may interfere with its dynamics, ultimately leading to reduced performance.
>
> ### 5. Sensitivity analysis
>
> We conduct a sensitivity analysis by fine-tuning the BERT-base model on the QNLI task from the GLUE benchmark using a batch size of 32 and a learning rate of 1.1e-4.
>
> To assess the impact of the mask interval parameter $m$, we disable the density scheduler and vary $m$ within the range [10, 1500]. The corresponding evaluation performance is presented in the table below. Our results show that model performance remains largely stable across this range, with optimal results observed when m falls between 80 and 300. Given that the total number of steps in one epoch is $S = 3274$ in our experiment, this tuning range corresponds to approximately $0.024S$ to $0.09S$. The hyperparameters reported in Table 10 of the manuscript are chosen to fall within this empirically validated range.
>
> | m        | 10    | 50    | 80    | 100   | 300   | 500   | 700   | 900   | 1100  | 1500  |
> | -------- | ----- | ----- | ----- | ----- | ----- | ----- | ----- | ----- | ----- | ----- |
> | Eval (%) | 90.72 | 90.85 | 90.99 | 91.21 | 91.21 | 91.09 | 91.09 | 91.16 | 91.03 | 90.98 |
>
> For the sensitivity analysis of the density interval $d$, we fix the mask interval at $m = 81$ and vary $d$ within the range [100, 1000]. The resulting evaluation performance is presented in the table below. We observe that when $d$ exceeds 300, the performance remains largely stable, with the best performance achieved at $d = 400$. Given that the total number of steps in one epoch is $S = 3274$, this corresponds to a density interval greater than approximately $(300/3274)S=0.092S$, with the optimal interval around $(400/3274)S=0.122S$. The hyperparameters reported in Table 10 of the manuscript follow this observation.
>
> | d        | 100   | 200   | 300   | 400   | 500   | 600   | 700   | 800   | 900   | 1000  |
> | -------- | ----- | ----- | ----- | ----- | ----- | ----- | ----- | ----- | ----- | ----- |
> | Eval (%) | 90.94 | 90.68 | 91.34 | 91.58 | 91.32 | 91.27 | 91.23 | 91.29 | 91.43 | 91.21 |
>
> ### 6. Scaling to larger models
>
> We appreciate the reviewer’s suggestion to evaluate our method on larger models. To address this, we conducted additional experiments on two larger-scale models. Specifically, we fine-tuned the LLaMA3.2-3B model on the Commonsense Benchmark, as described in our response to Question 2, and we fine-tuned the LLaMA 2-7B model on the GSM8K dataset.
>
> For the 7B model experiments, we used 4 random seeds and report the average performance along with standard deviation in the table below. The results demonstrate that our proposed method continues to offer performance gains when scaled to larger models.
>
> | method    | Eval acc     | Memory (GB) | Runtime (h) |
> | --------- | ------------ | ----------- | ----------- |
> | AdamW8b   | 33.74 (0.49) | 43.27       | 0.4         |
> | MicroAdam | 34.06 (0.66) | 38.9        | 0.47        |
> | NanoAdam  | 34.95 (0.64) | 36.68       | 0.41        |
>
> ### Limitations
>
> 1. To evaluate the potential limitations of our method when transferring across dissimilar domains, we fine-tuned a ResNet-18 model pretrained on ImageNet (a general-domain dataset) on the PathMNIST task from the MedMNIST dataset[1]—a medical image classification task with 9 classes. We compared the performance of full fine-tuning using AdamW (learning rate = 7e-4) with our method using NanoAdam (sparsity = 1%, learning rate = 1e-3). AdamW achieve evaluation accuracy 90.85% while NanoAdam achieves 90.63%. Despite the domain shift, NanoAdam achieves performance comparable to full fine-tuning, demonstrating its robustness even in cross-domain adaptation scenarios.
>
>     [1] Yang, Jiancheng, Rui Shi, and Bingbing Ni. "Medmnist classification decathlon: A lightweight automl benchmark for medical image analysis." 2021 IEEE 18th International Symposium on Biomedical Imaging (ISBI). IEEE, 2021.
>
> 2. To evaluate sensitivity to the initial weight distribution, we conducted the following experiment: we started with a ResNet-18 model pretrained on ImageNet and pruned 80% of the weights based on magnitude. We then fine-tuned the resulting sparse network on CIFAR-10 using both Adam (learning rate 1e-3) and NanoAdam (graident density=1%, learning rate 5e-3). Adam achieved an evaluation accuracy of 85.34%, while NanoAdam reached 86.16%, suggesting that NanoAdam is even more effective under sparse initialization. We hypothesize that this is because pruning removes 80% of the small weights, leaving the network to rely primarily on large weights during full fine-tuning—a strategy we have shown to be inefficient in our ablation study. In this case, focusing updates on the remaining small weights, as NanoAdam does, leads to better adaptation and generalization.

---

> > ### Comment · Reviewer_Vq6D · 2025-08-06
> >
> > Excellent rebuttal! and thanks for conducting appropriate experiments to address my concerns.

---

> > > ### Author Response · Authors · 2025-08-06
> > >
> > > Thank you so much for your feedback! I’m glad the additional experiments helped clarify things. I truly appreciate your engagement and the opportunity to strengthen the work through your insights.

---

### Official Review · Reviewer_S7zk · 2025-07-02

**Clarity:** 4
**Significance:** 2
**Originality:** 2
**Rating:** 5
**Confidence:** 3

**Summary:**

In this paper, the authors observe an inverse association between gradient magnitude and weight magnitude. They use this to motivate a finetuning strategy that focuses on weights with small magnitude, with a dynamically updated mask selecting those smallest weights ("NanoAdam"). They investigate this strategy theoretically in a teacher-student setup that illustrates the merits of this criterion. In empirical experiments on large pretrained models, they then find that NanoAdam improves both finetuning generalization and catastrophic forgetting compared to alternative methods while having a smaller memory footprint.

**Questions:**

1. Could you provide more detail for the proof of Theorem 2.2? Moreover, I think drawing a more explicit connection between Definition 2.1 and catastrophic forgetting (see my comments above) would be helpful. As part of that, it would also be useful to make Definition 2.1 more specific (e.g. how is the set of largest neurons selectedt? Is it the k largest neurons? Is it all neurons above a certain magnitude?

2. Could you clarify your contributions relative to previous work and, in particular, reference [18]? In particular, how central is the introduction of dynamic masking to your overall contributions?

3. If possible, could you add a comparison for the experiments in Section 4 to static masks as used in [18]?

4. Could you add standard deviations to the learning curves?

I can imagine that new experiments may not be realistic to run within the period of the rebuttal. Addressing the first two questions would be sufficient for me to increase my score though I believe addressing the latter two questions would substantially improve the paper further.

**Ethical Concerns:**

["NO or VERY MINOR ethics concerns only"]

**Final Justification:**

I thank the authors for their thorough response. The further clarification of the theoretical investigation is very helpful and the additional experiments support the authors' claims further. Finally, I found the authors' clarification about their novel contributions compared to prior work very helpful. Given that the authors addressed all of my questions, and with the understanding that they will update the camera-ready, I will increase my score to a 5.

**Limitations:**

yes

**Paper Formatting Concerns:**

No concerns.

**Quality:**

3

**Strengths And Weaknesses:**

**Strengths**

The paper was well-written and it was generally easy to follow the flow of logic. I also appreciated the fact that the authors integrated a theoretical and empirical investigation into the same phenomenon. Further, I thought the investigation into the association between gradient and weight magnitude was well-executed; I personally found this effect intriguing and I also appreciated the additional investigation into how this effect emerges over pretraining, as it helps shape the reader's intuition.

**Weaknesses**

My primary concerns with the paper in its current form are:

*1 Soundness of the theoretical investigation*

As noted above, I generally appreciated the theoretical investigation and thought it provided a useful simplified intuition (indeed, even as an empirical toy model, I think an approach like this is helpful). However, I think the authors should provide more detail on their setup. In particular, Theorem 2.2 does not provide sufficient detail: what does it mean that "gradient flow learns f_ft"? Is this in the limit of infinite data? How does this follow from [9] and which result are you referring to? The closest fit I could find was Theorem 2.3 in [9], but that talks about a relative comparison between iterative magnitude pruning and learning rate rewinding.

Two minor comments: I think it would be helpful to more concretely connect the "representation preserving" phenomenon to its resulting impact on catastrophic forgetting, i.e. how much worse does the network get at the pretrained task as a result of finetuning. At the moment, you only talk about the impact on the representation, but ultimately we care about the impact on model output. I think, with Lemma C.1, you should be able to make a statement about this, and I think that would help tie this argument together.

Furthermore, I don't understand the following statement: "Due to the implicit sparsity bias in overparameterized training, there often exists at least one sufficiently small neuron that has minimal impact on the pretrained representation." I agree that this is likely to happen in practice, but if I'm not missing anything, the implicit sparsity bias (if you're referring to works like [1-3]) usually refers to a sparsity in function space rather than weight space. I.e. you can cluster the neurons in a two-layer network into a sparse set of clusters but that doesn't mean the hidden layer is actually sparse. Could you elaborate on this point?

*2 Contributions of the empirical investigation*

In terms of your empirical investigation, am I understanding correctly that reference [18] already proposed and evaluated the idea to update small weights? If I'm understanding the relative contributions correctly, the primary methodological contribution you are introducing is to dynamically update this mask, right? I appreciated the comparison you drew between those approaches in Figure 11 and I think it would be important to evaluate the impact of dynamical masks more broadly in Section 4 as well, by specifically comparing the dynamic mask to the static mask across the entire set of benchmarks (extending your analysis in Figure 11). More generally, I think it would be good to contrast the novel contributions made in this paper to those already known from this previous paper in more detail. For example, are your insights into the benefits of (even static) masks for catastrophic forgetting new?

Additionally, I think it would be useful to repeat the experiments in Section 3.2 several times and indicate standard deviations of the different learning curves. It is difficult to assess how noisy these observations are.

**Conclusions**

Overall, in its current state, I rate this paper as a borderline reject. However, I enjoyed reading it, think it is a worthwhile study, and highlight a number of questions below that would cause me to increase my score.

**References**
1. https://proceedings.mlr.press/v125/chizat20a.html
2. https://proceedings.mlr.press/v99/savarese19a.html
3. https://proceedings.neurips.cc/paper_files/paper/2022/hash/7eeb9af3eb1f48e29c05e8dd3342b286-Abstract-Conference.html

---

> ### Author Rebuttal · Authors · 2025-07-30
>
> We appreciate the reviewer’s recognition of our work and that they found both our theory and empirical analysis useful. We thank them for their constructive and valuable feedback, which we address below. We look forward to a potential exchange during the discussion period.
>
> ### 1. Soundness of the theoretical investigation
> We are delighted that the reviewer found our toy model useful and are happy to answer their questions on the setup.
>
> **Details on theoretical setup**:  We consider the same setup as [1], which is a simpler setting than [9],  which considers the infinite data limit, i.e. learning over a given data distribution. We assume that the pre-training data has been generated from a teacher model $f_{\text{teacher}}$ consisting of $k$ neurons to define labels. On this data, first, a pre-trained model $f_{\text{pre}}$ is learned that consists of $n$ neurons, assuming $n>k$. The finetuning task is defined with labels from a target model that adds an extra neuron to the original teacher model $f_{\text{finetune}} = f_{\text{teacher}} + f_{\text{extra}}$. See also Appendix C for more details.
> This setup is equivalent to the finetuning scenario of training a single student neuron of the network $f_{\text{pre}}$ to approximate a single target neuron $f_{\text{extra}}$ by minimizing the mean squared error loss as average over general feature data, for instance $X \sim N(0,I_d)$ with $X \in \mathbb{R}^d$ with $d > 1$ (as in the illustration).
> Denote $a_{j}$ and $w_{j}$ as parameters of the trainable student neuron from our Theorem 2.2 and similarly denote $\tilde{a}$ and $\tilde{w}$ as parameters of the target neuron.
> Based on this notation and clarification, we would be happy to update Theorem 2.2 to the following more detailed formulation:
>
> Theorem 2.2:
> Assume a model $f(x)$ consisting of $n$ neurons learns the teacher $f_{\text{teacher}}(x)$ corresponding to a pre-training task so that $f(x) = f_{\text{teacher}}(x)$ for all $x\in \mathbb{R}^d$. Furthermore, let $f(x)$ consist of at least two neurons $i,r \in [n]$ such that $\max(\{|a_i|^2, |a_r|^2\}) \leq \epsilon$ for an $\epsilon > 0$ and $sign(a_i) \neq sign(a_r)$. Let a new task be defined based on labels $f_{\text{finetune}} :=  f_{\text{teacher}} + f_{\text{extra}}$ with an extra neuron $f_{\text{extra}} = \tilde{a} \sigma( \tilde{w} \cdot)$. Let only the neuron $j$ of $f$ be trainable to finetune $f(x)$ to the new task, where $j=arg\min_{i}{\{|a_i|||w_i||: sign(a_i)=sign(\tilde{a})\}}$. Then, the gradient flow with respect to finetuning time $t$ of the neuron $j$ parameterized as $v_{j,t} = |a_{j,t}| w_{j,t}$ and initialized at pre-trained values $v_{j,0}=|a_j| {w_j}$, converges to a value $v_{\infty}$ so that $|v_{\infty} - v| <\epsilon$, where $v$ is the target $v = |\tilde{a}| \tilde{w}$.
>
>
> **Details on the assumptions and proof idea**: Note that the assumption $\max(\{|a_i|^2, |a_r|^2\}) \leq \epsilon$ in the above theorem is justified because $f$ learns a representation of  $f_{\text{teacher}}(x)$ when solving a pre-training task where at least one of the neurons is effectively pushed to $0$ (as the teacher consists also of fewer neurons than the trained model $f$).
>
> To apply Theorem 6.4 in [1], we need matching signs for the parameters $a$. Otherwise, we have an immediate mismatch between the two single neuron functions. Assuming matching signs, we can absorb $a_{j}$ and $\tilde{a}$ into the activation to simplify the analysis. This reduces the problem to optimizing a single layer neuron $\sigma(v_{j,t} \cdot)$ with initialization $|a_{j}| w_{j}$ to learn a target vector $v = |\tilde{a}| \tilde{w}$ with some small label perturbation that is equal to the initialization $a_j\sigma(w_j x_i)$.
> To prove this, we make some assumptions:  (1) the teacher representation has been exactly learned as in Figure 3 (2) the data set fulfills the assumptions from Theorem 6.4 (which cover  i.i.d. Gaussians).
> For now we assume there exists a neuron such that $|a|^2 \leq \epsilon$ after pretraining which likely exists after pretraining.
> Then, this noise term vanishes due to a concentration inequality on the parameter initialization, as we have $|a_{j}|^2 =||w_{j}||^2$. In summary, we can apply Theorem 6.4 to show that the gradient flow for $f$ with one trainable neuron learns $f_{\text{finetune}}$.
> We will highlight these details in the revised manuscript. Additionally, we will include the gradient flow of the one neuron problem in the main manuscript to explain the setting in more detail.
>
> [1] Yehudai, Gilad, and Shamir Ohad. "Learning a single neuron with gradient methods." Conference on Learning Theory. PMLR, 2020.
>
> **Definition 2.1 and catastrophic forgetting**: We will make Definition 2.1 more explicit using the notation of Theorem 2.2, as follows.
>
> **Definition 2.1**: A finetuned network $f$ is $k$-neuron representation preserving iff the largest neurons corresponding to $\text{Top-k}(j \in [n] : |a_j| ||w_j||)$ remain unchanged compared to the pre-training task.
>
>
> Definition 2.1 is directly connected to catastrophic forgetting, as the top-k neurons correspond to the part of the pre-training task that transfers to the new task. For that reason, we do not want to forget them but use them also in the finetuning task.
> This is also reflected in our $L_2$ proxy measure of parameter shift for catastrophic forgetting (as in Lemma C.1). Not only do the parameters of the top-k neurons not change during finetuning, they therefore do not contribute to the $L_2$ distance between old and new parameters. Also the other parameters have to change less. If some of the largest neurons would change, to compensate for that change, all trainable neurons would have to move more in parameter space. As a consequence, Definition 2.1 also implies low forgetting according to Lemma C.1, which we will also demonstrate numerically in our example in the appendix.
>
> **Implicit sparsity bias**: The implicit sparsity bias refers indeed to [2]. It implies that a product of parameters (in our case $|a|||w|| < \epsilon$) remains small. The product tends to be small if both or either one of the parameters is small. Products of parameters thus intuitively promote sparsity. There is a large line of work on this type of bias. One could potentially interpret the product as a form of function space, but this is beyond the scope of this paper and we will clarify our meaning of sparsity in our revised manuscript. We primarily want to argue why the assumption  $\max(\{|a_i|^2, |a_r|^2\}) \leq \epsilon$ in our theorem is justified (see comment above).
>
> [2] Arora, Sanjeev, et al. "Implicit regularization in deep matrix factorization." Advances in neural information processing systems 32 (2019).
>
>
> ### 2. Contributions of the empirical investigation
>
> We appreciate the opportunity to clarify our contributions relative to [18]:
>
> - While [18] hypothesizes that updating only small-magnitude weights can be effective, we provide an explanation: large gradients are usually associated with small weights during finetuning, making them more adaptable. We show that the gradient-weight correlation is much stronger during fine-tuning than training from scratch, explaining why the strategy is less effective in the latter case.
> - We attribute this pattern to knowledge transferability and overparameterization, offering a principled understanding of when small-weight finetuning is most effective.
> - We further show that updating small weights helps mitigate catastrophic forgetting, supported by both theoretical and empirical evidence.
> - Our method is novel in that: (a) mask selection and (b) sparsity are dynamic, and (c) selection is done per layer rather than globally. Unlike [18], which uses a fixed global mask, our dynamic approach improves efficiency and performance while reducing memory usage. Dynamic masking specifically improves the implicit weight regularization by discouraging over-reliance on a fixed subset of parameters; and mitigates catastrophic forgetting by reducing parameter shift.
> - Lastly, unlike [18] which focuses only on NLP, we evaluate our method on CV tasks as well, demonstrating broader applicability.
>
> ### 3. Comparison with static mask
> The additionally requested table below shows that our dynamic masking strategy (NanoAdam) leads to significant improvements compared to static masking [18].
>
> | Model       | Method              | COLA  | SST2  | MRPC  | STSB  | QQP   | MNLI  | QNLI  | AVG   |
> |-------------|---------------------|-------|-------|-------|-------|-------|-------|-------|--------|
> | bert-base   | NanoAdam            | 60.87 | 93.46 | 88.48 | 89.98 | 90.67 | 84.30 | 91.76 | 85.65  |
> | bert-base   | NanoAdam (static)   | 56.24 | 91.51 | 84.07 | 89.68 | 89.75 | 82.59 | 90.92 | 83.54  |
> | bert-large  | NanoAdam            | 66.85 | 94.61 | 90.20 | 90.86 | 91.03 | 86.40 | 92.44 | 87.48  |
> | bert-large  | NanoAdam (static)   | 59.07 | 92.66 | 84.31 | 89.86 | 90.33 | 85.39 | 91.78 | 84.77  |
> | opt-1.3B    | NanoAdam            | 67.69 | 96.45 | 87.99 | 91.00 | 91.33 | 88.24 | 92.75 | 87.92  |
> | opt-1.3B    | NanoAdam (static)   | 60.38 | 95.30 | 86.03 | 90.55 | 91.07 | 87.61 | 91.91 | 86.12  |
>
> ### 4. Standard deviations
> We will add confidence intervals to our figures and tables based on five seeds. Please find exemplary results below.
>
> | Method         | Averaged Performance | Standard Deviation |
> |----------------|----------------------|---------------------|
> | Small weights  | 89.4                 | 0.296               |
> | Large weights  | 62.25                | 0.927               |
> | Random weights | 88.88                | 0.229               |

---

> > ### Comment · Reviewer_S7zk · 2025-08-01
> >
> > I thank the authors for their thorough response. The further clarification of the theoretical investigation is very helpful and the additional experiments support the authors' claims further. Finally, I found the authors' clarification about their novel contributions compared to prior work very helpful. Given that the authors addressed all of my questions, and with the understanding that they will update the camera-ready, I will increase my score to a 5.

---

> > > ### Author Response · Authors · 2025-08-04
> > >
> > > We sincerely thank the reviewer for their thoughtful engagement and for taking the time to reassess our work. We are glad that our clarifications are helpful in addressing your concerns. We appreciate your constructive feedback throughout the review process, which has strengthened our paper.

---

### Official Review · Reviewer_tQEr · 2025-07-05

**Clarity:** 3
**Significance:** 3
**Originality:** 2
**Rating:** 4
**Confidence:** 3

**Summary:**

The paper presents a unique finding that during fine-tuning of large models, the weights with small magnitude often get large gradients. Based on this observation, the paper proposes a modified Adam optimizer to dynamically update only the small-magnitude weights during fine-tuning in order to reduce the cost of fine-tuning large pretrained neural networks. The paper presents results to support this method on both NLP and vision tasks.

**Questions:**

The writing could have been better organized.
Due to the significant reduction in memory usage and updatable parameters, is the proposed method also prone to overfitting?

**Ethical Concerns:**

["NO or VERY MINOR ethics concerns only"]

**Final Justification:**

The rebuttal addresses my concerns. It is an interesting work, but considering the scope of the work is restricted to fine-tuning, I would like to retain my initial rating.

**Limitations:**

Yes

**Quality:**

3

**Strengths And Weaknesses:**

Strengths:
The paper takes an interesting approach to restrict the fine-tuning to parameters with small absolute magnitude.
Demonstrates superior or equivalent performance with lower memory usage.


Weaknesses:
It can be used for fine-tuning and not for training from scratch, since the core hypothesis will not hold true.
No discussion on overfitting vulnerability due to the reduction in trainable parameters.

---

> ### Author Rebuttal · Authors · 2025-07-30
>
> We thank the reviewer for finding our approach interesting and acknowledging the performance gains and memory reductions of our method. Below, we address their insightful comments and look forward to the discussion period.
>
> Regarding the concern about potential overfitting, we would like to clarify the following:  In fact, our proposed method is inherently less prone to overfitting. As demonstrated in Table 5 and Appendix G.3, our method results in a smaller parameter shift away from the initial weights, indicating that it learns the downstream task with smaller movement in parameter space, which also mitigates catastrophic forgetting of the pre-training task.
>
> Another perspective is that our method implicitly performs a form of regularization, similar to weight decay. By selectively updating only a subset of parameters—specifically avoiding those with large magnitudes—we prevent dominant features from disproportionately shaping the network’s functional space. This restraint helps mitigate overfitting and promotes more robust generalization.
>
> Indeed, our method primarily concerns finetuning, which is a common and highly relevant problem in the era of large models. Due to the simplicity of our parameter subset selection criterion, its advantages, including lower memory requirements, faster convergence, and improved generalization, can be easily realized across different architectures and training setups.

---

> ### Author Response · Authors · 2025-08-07
>
> We thank you for your time and effort in reviewing our paper. As the author–reviewer discussion deadline is fast approaching, we kindly request your response to our rebuttal at your earliest convenience.
> We would be happy to answer any open questions upon request.

---

> > ### Comment · Reviewer_tQEr · 2025-08-08
> >
> > The rebuttal addresses my concerns.

---

### Decision · Program_Chairs · 2025-09-17

**Decision:**

Accept (poster)

**Comment:**

The paper analyses the correlation of small weights and gradients and identifies that they are correlated - therefore they propose an efficient adam variant which only updates small weights in stead of all. Leading to less forgetting as well as improved performance.

Strengths: Paper provides nice empirical evidence for their claims, method is interesting, and paper is easy to follow and nicely structured.

Weaknesses: [18] already proposes a very similar method, and this paper only proposes small but potentially important changes. Contributions are not entirely clear, and the rebuttal by the authors is not totally satisfactory in my opinion. Nevertheless, the reviews are all positive and mostly concerns were adressed during the rebuttal.

This is definitely a borderline paper for me, although I lean towards acceptance.